# Post-ejaculatory inhibition of female sexual drive via heterogeneous neuronal ensembles in the medial preoptic area

Kentaro K Ishii[1,2], Koichi Hashikawa[1,2], Jane Chea[1,2], Shihan Yin[1,2], Rebecca Erin Fox[1,2], Suyang Kan[1,2], Meha Shah[1,2], Zhe Charles Zhou[1,2], Jovana Navarrete[1,3], Alexandria D Murry[1,3], Eric R Szelenyi[1,3], Sam A Golden[1,3], Garret D Stuber[1,2,4]*

[1]Center for the Neurobiology of Addiction, Pain, and Emotion, University of Washington, Seattle, United States; [2]Department of Anesthesiology and Pain Medicine, University of Washington, Seattle, United States; [3]Department of Biological Structure, University of Washington, Seattle, United States; [4]Department of Pharmacology, University of Washington, Seattle, United States

*For correspondence:
gstuber@uw.edu

Competing interest: The authors declare that no competing interests exist.

## eLife Assessment

This **important** work combines molecular genetics and behavioral analyses to identify inhibitory neurons in the female medial preoptic area as a neural locus that is activated following male ejaculation and whose prolonged activity plays a key role in the regulation of female sexual motivation. These experiments are rigorous and well-performed. The data are **compelling** and demonstrate that a subpopulation of neurons in the medial preoptic area are selectively activated following the completion of mating in females. The medial preoptic area has long been implicated as critical to sexual behavior in both sexes; however the use of a self-paced mating assay for females provides fine control over manipulating and monitoring cellular activity in this region during more naturalistic behavior. In addition, this study may act to inspire others to further explore the additional brain regions found to show upregulation of neural activity (Fos) during mating completion in females using the datasets generated here.

**Abstract** Male ejaculation acutely suppresses sexual motivation in male mice. In contrast, relatively little is known about how male ejaculation affects sexual motivation and sexual behavior in female mice. How the brain responds to the completion of mating is also unclear. Here, by using a self-paced mating assay, we first demonstrate that female mice show decreased sexual motivation acutely after experiencing male ejaculation. By using brain-wide analysis of activity-dependent labeling, we next pin-pointed the medial preoptic area as a brain region strongly activated during the post-ejaculatory period. Furthermore, using freely moving *in vivo* calcium imaging to compare the neural activity of inhibitory and excitatory neurons in the medial preoptic area, we revealed that a subset of the neurons in this region responds significantly and specifically to male ejaculation but not to female-to-male sniffing or to male mounting. While there were excitatory and inhibitory neurons that showed increased response to male ejaculation, the response magnitude as well as the proportion of neurons responding to the event was significantly larger in the inhibitory neuron population. Next, by unbiased classification of their responses, we also found a subpopulation of neurons that increase their activity late after the onset of male ejaculation. These neurons were all inhibitory indicating that male ejaculation induces a prolonged inhibitory activity in the medial preoptic area. Lastly, we found that chemogenetic activation of medial preoptic area neurons that

were active during the post-ejaculatory period, but not during appetitive or consummatory periods, were sufficient to suppress female sexual motivation. Together, our data illuminate the importance of the medial preoptic area as a brain node which encodes a negative signal that sustains a low sexual motivation state after the female mice experience ejaculation.

## Introduction

Sexual behavior is a fundamental and universal component of the observable behavioral spectrum in all mammalian species. This behavior results in reproduction, the vital process for species propagation. However, reproduction carries significant burdens, particularly for females, spanning pregnancy, birth, nursing, offspring rearing, and heightened vulnerability to environmental threats. Consequently, the female ability to strictly control sexual drive is adaptive and confers individual survivability. This can involve both increasing and decreasing motivation to engage in sexual behavior, dependent on anticipated beneficial outcomes. Rodents provide a powerful resource for studying female sexual behavior and the underlying neural basis (*Agrati, 2022*; *Micevych and Meisel, 2017*). Typically, rodent sexual behavior is divided into three periods: an appetitive, consummatory, and post-ejaculatory phase (*Hashikawa et al., 2016*; *Ishii and Touhara, 2019*; *Lenschow and Lima, 2020*). Importantly, in males, sexual motivation acutely reduces after ejaculation (*Hull et al., 2005*). However, less is known about how the experience of male ejaculation in female mice impacts their sexual behavior and motivation. Furthermore, how the transient experience of male ejaculation is represented in the brain to change the behavioral outcome is not well understood. The medial preoptic area (MPOA), a molecularly and functionally heterogenous brain region, has classically been known to regulate wide variety of social behaviors (*Fang et al., 2018*; *Hashikawa et al., 2021*; *Karigo et al., 2021*; *Kohl et al., 2018*; *Moffitt et al., 2018*; *Tsuneoka and Funato, 2021*; *Wei et al., 2018*; *Paredes, 2003*). Indeed, the MPOA has also been described to be an important brain area for the regulation of female sexual behavior. Early studies using histological approaches to map neural activity in rats have also shown MPOA, together with other hypothalamic regions, to be activated during sexual behavior, after male ejaculation, and after genital stimulations (*Parada et al., 2010*; *Pfaus et al., 1996*). Studies in female rats using electrophysiological recording or histological approaches have reported that the MPOA contains neurons with various response specificity to sexual appetitive behaviors such as sniffing, consummatory behaviors such as male mounting, intromission, and ejaculation (*Kato and Sakuma, 2000*; *Pfaus et al., 1993*). Similarly, MPOA lesions increased sexual receptivity, suggesting that the MPOA negatively regulates female sexual behavior when intact (*Powers and Valenstein, 1972*). Interestingly, the MPOA has also been found to positively regulate female sexual appetitive behavior through their projection to the mesolimbic reward system (*McHenry et al., 2017*). However, how the MPOA both positively and negatively regulates female sexual behavior, furthermore, sexual motivation, has been a long-lasting question in the field. Here, we first assess sexual motivation in female mice by quantifying a measurable approach or avoidance of a male conspecific. Next, by combining cellular-resolution whole-brain mapping of activity-dependent labeling, *in vivo* calcium imaging, and chemogenetic neural manipulation, we find that a subset of neurons in the MPOA, which are largely inhibitory, respond to male ejaculation but not to sexual appetitive behavior. This population is sufficient to suppress female sexual motivation. Our findings illuminate the importance of the MPOA as a brain region that encodes a negative signal that sustains low sexual motivation following male ejaculation in female mice.

## Results

### Female sexual motivation is suppressed after male ejaculation

Experimentally, sexual motivation can be quantified by the frequency of a subject displaying appetitive behavior, such as approach and sniffing behavior, toward an opposite-sex conspecific. However, in a traditional rodent mating assay, where the female and male both freely interact with each other, it is difficult to isolate female-initiated appetitive behavior, and thus their sexual motivation. To directly address aspects of female sexual motivation independent from the male's behavior, we used a female self-paced mating assay (*Erskine, 1989*; *Erskine, 1985*; *Peirce and Nuttall, 1961*). In this procedure, a female subject and a sexually experienced male are placed in a behavior apparatus divided by a wall

with a hole small enough for a female to move through, but not the male. The female subject may move into the area where the male animal was placed ('Interaction zone') or move out to the other area ('Isolation zone') (*Figure 1A*). Thus, in this assay, the female has direct control of when interaction happens and provides a more ethologically valid measure of female appetitive sexual behavior. Specifically, the amount of time the female subject spends in the interaction zone and the number of transitions they make serve as metrics to quantify female appetitive behavior. On the experiment day, the female subject, which was primed with estradiol to be behaviorally estrus, was placed in the apparatus with the partner ('Sexual interaction' trial) or alone ('Control' trial). Each animal went through both trials on different experiment days. The position of the animal was tracked to quantify the time spent in each zone. First, we found that the virgin female mice stayed in the isolation zone significantly longer than sexually experienced females, suggesting that previous sexual experience increased sexual motivation in the female mice (*Figure 1—figure supplement 1*). Next, in the sexually experienced animals, we observed robust behavioral changes after male ejaculation. First, we observed that female subjects tended to spend more time in the isolation zone after male ejaculation compared to post- sniffing, mounting, and intromission (*Figure 1B–F*). To quantify the changes in animal behavior, we divided the sexual interaction trial into pre- and post- male ejaculation epochs (*Figure 1G–M*). As a result, we found that there was a significant decrease in the number of zone transitions as well as a significant increase in the time spent in the isolation zone after male ejaculation. Importantly, this change was not dependent on the state of the male animal (*Figure 1—figure supplement 2*). When the female animal was introduced to a novel sexually motivated male animal immediately after they experienced male ejaculation from another mouse, they showed a similar reduction in zone transition and an increase in the time spent in the isolation zone. To further evaluate the female animal's motivation to engage in sexual behavior, we quantified the latency to return to the interaction zone after a behavioral episode (*Bermant, 1961*; *Bermant and Westbrook, 1966*). This was the largest after the female experienced male ejaculation when compared to after sniffing or after mounting (*Figure 1I*). In addition, while there was no change in the amount of female-to-male sniffing, there was a significant decrease in the number of mounting and intromission episodes (*Figure 1J–L*). There was also a significant increase in the number of self-grooming behavioral episodes (*Figure 1M*). Animals that were in their natural estrus cycle showed similar behavior changes after experiencing male ejaculation (*Figure 1N–T*). Interestingly, animals that were in their natural estrus cycle showed a strong reduction of sniffing after experiencing male ejaculation but no changes in self-grooming behaviors (*Figure 1Q and T*). Collectively, these results suggest that male ejaculation acutely decreases sexual motivation.

## Brain-wide mapping of neural response to male ejaculation using activity-dependent labeling in the female brain

We next questioned how male ejaculation suppresses female sexual motivation. To investigate this question, we first determined which brain regions were associated with neural activity during the post-ejaculatory period in an unbiased manner. We conducted a cellular-resolution brain wide activity mapping analysis using targeted recombination in active populations method (TRAP), which allows genetic labeling and access to neurons that are active during a specific time window determined by tamoxifen administration (*DeNardo et al., 2019*). To capture the activity pattern, Fos-2A-iCreER$^{T2}$::RCL-tdTomato (TRAP2::Ai14) female mice were administered 4-hydroxytamoxifen (4-OHT) immediately after male ejaculation ('post-ejaculatory' group) (*Figure 2A*). In this method, cells that show behaviorally induced Fos activity were labeled with tdTomato. In addition to these animals, we also prepared a group of animals that were administered with 4-OHT while the animals were in the appetitive phase or in the consummatory phase. For the appetitive group, female subjects interacted with male mice and displayed female-to-male sniffing but did not experience any mount or intromission behavior, nor male ejaculation. For the consummatory group, female subjects interacted with male mice and displayed female-to-male sniffing, experienced mount or intromission behavior, but not male ejaculation. After 2 weeks, the tissue was collected, cleared, imaged, and processed for registration on a template brain atlas (*Park et al., 2019*; *Renier et al., 2016*; *Figure 2A*). The tdTomato+ cells were segmented using a trained classifier and normalized by the area of each brain region (*Figure 2B*). As a result, we found the level of tdTomato expression increased in a brain-wide manner during the post-ejaculatory period when compared to the appetitive phase (*Figure 2C–D*, *Supplementary file 2*). We further noticed that regions well known to be related to social behavior

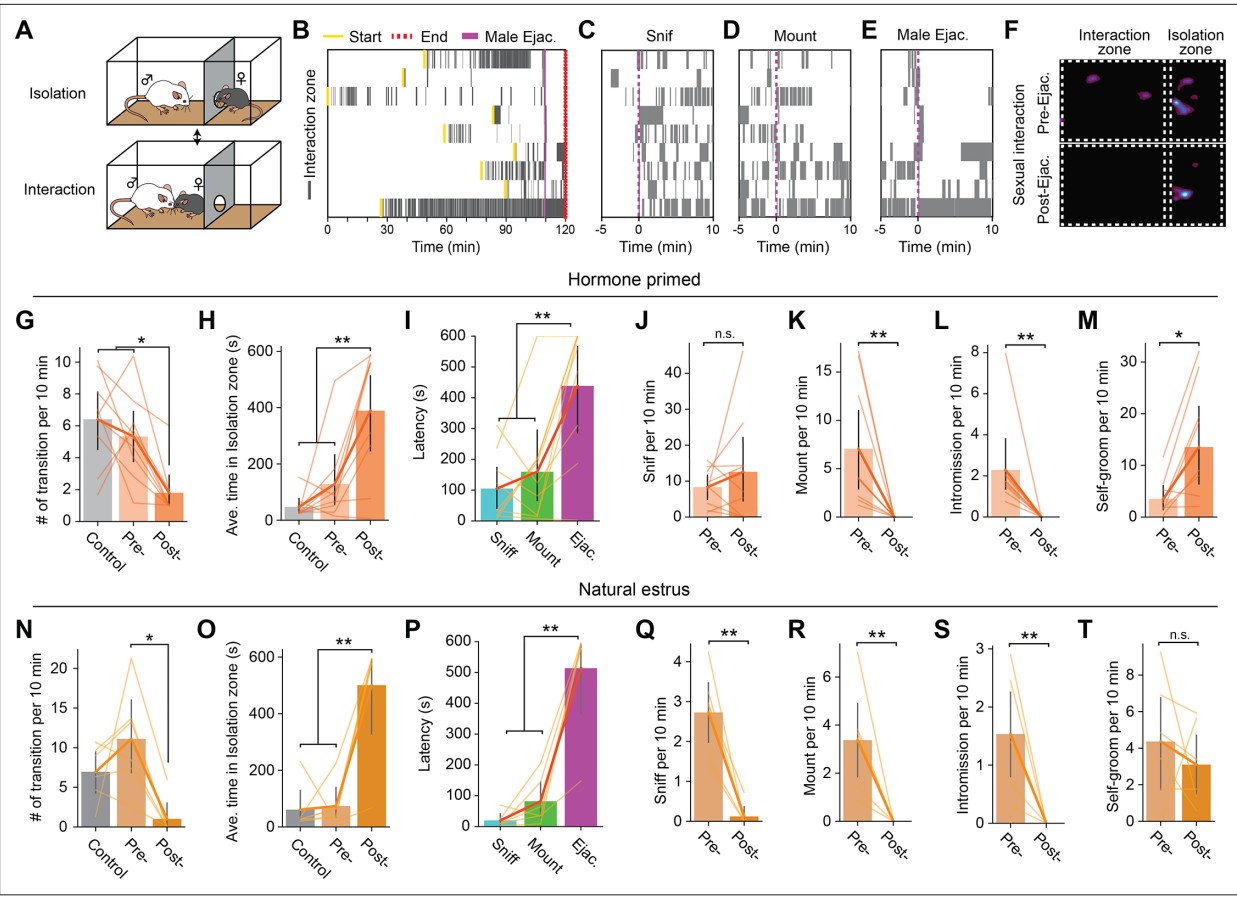

**Figure 1.** Female mice show decreased sexual motivation after male ejaculation. (**A**) Schematic of female self-paced mating assay. The behavior apparatus was divided by a wall with a hole small enough for only the female subject to go through. The female subject was allowed to freely choose between 'interaction' with the male partner in the larger zone or 'isolation' in the smaller zone. Results using ovariectomized and hormone primed female mice are shown in **B–M**. Results using naturally estrus female mice are shown in **N–T**. (**B**) Raster plots of time spent in the interaction zone (gray). The onset of male ejaculation, start and the endpoint of the experiment is shown in yellow, blue, and red, respectively. (**C–E**) Raster plots of time spent in the interaction zone (gray) around the onset of the first female-to-male sniffing, male mounting, intromission and male ejaculation. (**F**) Representative heatmap showing the time spent in the behavior apparatus during a control trial or a sexual interaction trial. The sexual interaction trial was further divided into pre- and post- male ejaculation. (**G–H**) Behavior analysis for hormone primed animals. (**G**) Number of transitions between the zones per 10 min. RM ANOVA; F(2,16)=10.89, **p=0.001035889. Tukey's HSD; Control vs Post, **p=0.0015. Control vs Pre, p=0.6284. Post vs Pre, *p=0.0142. (**H**) Average time spent in the isolation zone during the control trial, and pre- and post- male ejaculation. RM ANOVA; F(2,16)=14.34, **p=0.00027. Control vs Post, ***p=0.0004. Control vs Pre, p=0.5361. Post vs Pre, **p=0.0057. (**I**) Latency to return to the interaction zone after sniff, male mount or male ejaculation. RM ANOVA; F(2,16)=11.87, **p=0.00069052. Ejaculation vs Sniff, **p=0.0016. Ejaculation vs Mount, **p=0.0077. Sniff vs Mount, p=0.7951. (**J**) Number of female-to-male sniffing per 10 min during pre- and post- male ejaculation. p=0.65234375. (**K**) Number of male mounting per 10 min during pre- and post- male ejaculation. **p=0.00390625. (**L**) Number of intromission per 10 min during pre- and post- male ejaculation. **p=0.00390625. (**M**) Number of female self-grooming per 10 min during pre- and post- male ejaculation. *p=0.025061844. (**N–T**) Behavior analysis for naturally estrus animals. (**N**) Number of transitions between the zones per 10 min. RM ANOVA; F(2,10) = 7.54, *p=0.01007. Tukey's HSD; Control vs Post, p=0.0871. Control vs Pre, p=0.2727. Post vs Pre, **p=0.0038. (**O**) Average time spent in the isolation zone during the control trial, and pre- and post- male ejaculation. RM ANOVA; F(2,10) = 15.71, ***p=0.00082. Control vs Post, ***p=0.0002. Control vs Pre, p=0.9862. Post vs Pre, ***p=0.0002. (**P**) Latency to return to the interaction zone after sniff, male mount, or male ejaculation. RM ANOVA; F(2,10) = 38.36, ***p=0.00002. Ejaculation vs Sniff, ***p=0. Ejaculation vs Mount, ***p=0. Sniff vs Mount, p=0.6201. (**Q**) Number of female-to-male sniffing per 10 min during pre- and post- male ejaculation. *p=0.03125. (**R**) Number of male mounting per 10 min during pre- and post- male ejaculation. *p=0.03125. (**S**) Number of intromission per 10 min during pre- and post- male ejaculation. *p=0.03125. (**T**) Number of female self-grooming per 10 min during pre- and post- male ejaculation. p=0.3452. All data are shown as mean ± 95% confidence interval and were analyzed by RM ANOVA with Tukey HSD post-hoc test (FWER = 0.05) (**G–I** and **N–P**) or Wilcoxon signed-rank test (**J–M** and **Q–T**). n=9 (**D–K**), n=6 (**N–T**). *p<0.05, **p<0.01, ***p<0.001; ns, not significant.

The online version of this article includes the following figure supplement(s) for figure 1:

**Figure supplement 1.** Sexual experience increase sexual motivation in female mice.

**Figure supplement 2.** Reduction of sexual motivation after male ejaculation in female mice is not dependent on the state of the male mouse.

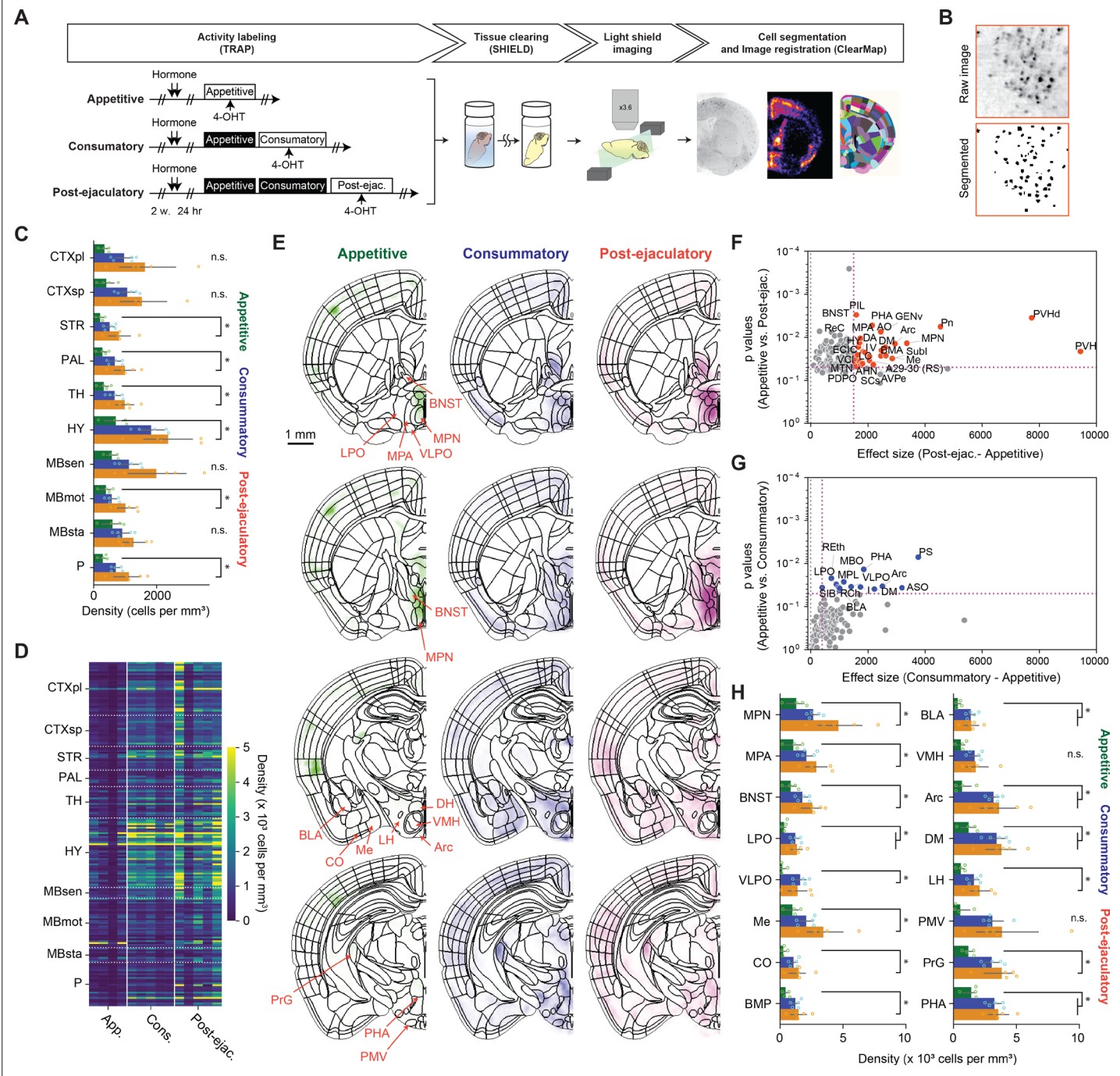

**Figure 2.** Brain-wide analysis of cells responding to male ejaculation in the female brain. (**A**) Overview of the experiment pipeline. After the activity labeling using targeted recombination in active populations method (TRAP), brain tissue was collected and cleared using SHIELD. The cleared tissue was imaged using a light-sheet microscope. The dataset was registered to a reference brain atlas and the number of tdTomato + cells were quantified using ClearMap and ilastik. (**B**) Representative image showing cell segmentation. (**C**) Density of tdTomato + cells per mm³ in larger brain regions. CTXpl: Cortical plate, CTXsp: Cortical subplate, STR: Striatum, PAL: Pallidum, VERM: Vermal regions, HEM: Hemispheric regions, TH: Thalamus, HY: Hypothalamus, MBsen: Midbrain, sensory related, MBmot: Midbrain, motor-related, MBsta: Midbrain, behavioral state-related, P: Pons. (**D**) Heatmap showing density of tdTomato + cells per mm³ in subregions. Each column indicates data from one subject. Each row indicates data from one subregion. (**E**) Heatmap of average tdTomato density for each group from representative coronal planes. Scale bar, 1 mm. (**F**) Scatterplot showing difference of average tdTomato density and p-value after Tukey's HSD post-hoc test between the post-ejaculatory group and the appetitive group. (**G**) Scatterplot showing difference of average tdTomato density and p-value after Tukey's HSD post-hoc test between the consummatory group and the appetitive group. (**H**) Density of tdTomato + cells per mm³ in a subset of subregions. The full list of brain regions is shown in *Figure 1—figure supplement 2*.

*Figure 2 continued on next page*

*Figure 2 continued*

MPN: Medial preoptic nucleus, MPA: Medial preoptic area, BNST: Bed nuclei of the stria terminalis, LPO: Lateral preoptic area, VLPO: Ventrolateral preoptic nucleus, Me: Medial amygdaloid nucleus, CO: cortical amygdaloid nucleus, BMP: Basomedial amygdaloid nucleus, posterior part, BLA: Basolateral amygdaloid nucleus, anterior part, BLP: Basolateral amygdaloid nucleus, posterior part, VMH: Ventromedial hypothalamic nucleus, Arc: Arcuate hypothalamic nucleus, DM: Dorsomedial hypothalamic nucleus, LH: Lateral hypothalamic area, PMV: Premamillary nucleus, ventral part, PrG: Pregeniculate nucleus of the prethalamus, PHA: Posterior hypothalamic area. All data are shown as mean ± 95% confidence interval and were analyzed by ANOVA test with Tukey's HSD post-hoc test (FWER = 0.05) (**C**): FDR = 0.05, (**I**): FDR = 0.10. Detailed statistical values are shown in *Supplementary file 2* and *Supplementary file 3*. Appetitive (green), n=4. Consummatory (blue), n=5. Post-ejaculatory (orange), n=5. *p<0.05, ns, not significant.

The online version of this article includes the following figure supplement(s) for figure 2:

**Figure supplement 1.** Brain-wide analysis of male ejaculation-responding cells in the female brain.

**Figure supplement 2.** Brain-wide analysis of neurons active during post-ejaculatory period in the female brain.

had significantly higher tdTomato density in the post-ejaculatory group compared to appetitive group (*Figure 2E–I*, *Figure 2—figure supplements 1 and 2*, *Supplementary file 3*). This includes paraventricular hypothalamic areas (PVHd, PVH, PaV) which are important for endocrinological regulations, medial amygdala (MeA) which is critical for olfactory sensory information processing, and the BNST (*Flanigan and Kash, 2022*; *Marsh et al., 2021*; *Raam and Hong, 2021*). The medial preoptic area nucleus (MPN), a subregion of the MPOA, had significantly higher tdTomato density in the post-ejaculatory group compared to the appetitive group. The MPN also had the fourth largest difference of density between the post-ejaculatory group and appetitive group among all the brain regions (*Figure 2F*). While it was not statistically significant, the consummatory group also tended to have a larger tdTomato density than the appetitive group in the MPN as well (*Figure 2G and H*). From this result, we decided to further investigate the role of MPOA and how it is related to the regulation of female sexual motivation after experiencing male ejaculation.

## Male ejaculation is represented by both inhibitory and excitatory cell types in the MPOA

MPOA is a molecularly heterogenous brain region. The area is a mixture of 80% inhibitory neurons and 20% of excitatory neurons which can be further classified into neuronal subtypes by marker gene expressions (*Hashikawa et al., 2021*; *Moffitt et al., 2018*). These subtypes are often associated with a biological function. One example is the *Nts*-expressing cells, which also tend to co-express *Esr1* and Gamma-aminobutyric acid (GABA) Transporter (Vgat) or *Slc32A1* (*McHenry et al., 2017*). In female mice, these neurons were shown to respond to male odor when the subject was sexually receptive, and the activation of these neurons was sufficient to increase sexual appetitive behaviors. Therefore, identifying the specific cell types which respond to male ejaculation but not to appetitive behavior is critical to understanding how the MPOA complex regulates female sexual motivation. Here, we utilized multiplexed *in situ* RNA hybridization chain reaction (HCR) to identify previously characterized cell type markers of MPOA and an immediate early gene *Fos* (*Figure 3A*; *Choi et al., 2014*). Brain samples were collected after the subject experienced male ejaculation ('post-ejaculatory' group) or during the appetitive phase; while the subject was interacting with the male mice without any consummatory behavior ('Appetitive' group) (*Figure 3B*). First, to understand which subregion of the preoptic area was activated, we quantified the number of *Fos+* cells. As a result, we found that the number of *Fos+* cells in the post-ejaculatory group was significantly higher than the appetitive group in the MPN but not in the nearby regions (*Figure 3C and D*). In the MPN, the amount of Vesicular glutamate transporter 2 (Vglut2) expressing cells Vglut2 and Vgat cells expressing *Fos* were significantly higher in the post-ejaculatory group than in the appetitive group, suggesting that male ejaculation activates both excitatory and inhibitory neurons (*Figure 3E–I*). The proportion of *Fos+* cells was higher in the Vgat population. To further investigate the cell types in the MPN that were activated by male ejaculation, we analyzed the co-expression of *Fos* and the MPOA cell type markers (*Figure 3J*). As a result, in the Vglut2 cells, we found that *Fos* induction was only observed in the cells that co-expressed *Calcr*. In contrast, in the Vgat-expressing cells, *Fos* induction was observed in *Calcr*, *Gal* and *Prlr* co-expressing cells. Next, we conducted clustering on the gene expression matrix to classify cells and analyzed the distribution of *Fos* among the clusters (*Figure 3K–M*). As a result, we were able to identify six potential cell types which we labeled as type 1: Vglut2, type 2: Vgat, type 3: *Gal*, type 4: *Nts*, type 5: *Calcr*, and type 6: NA, or a cell group that could not be categorized by a specific gene expression pattern. Type

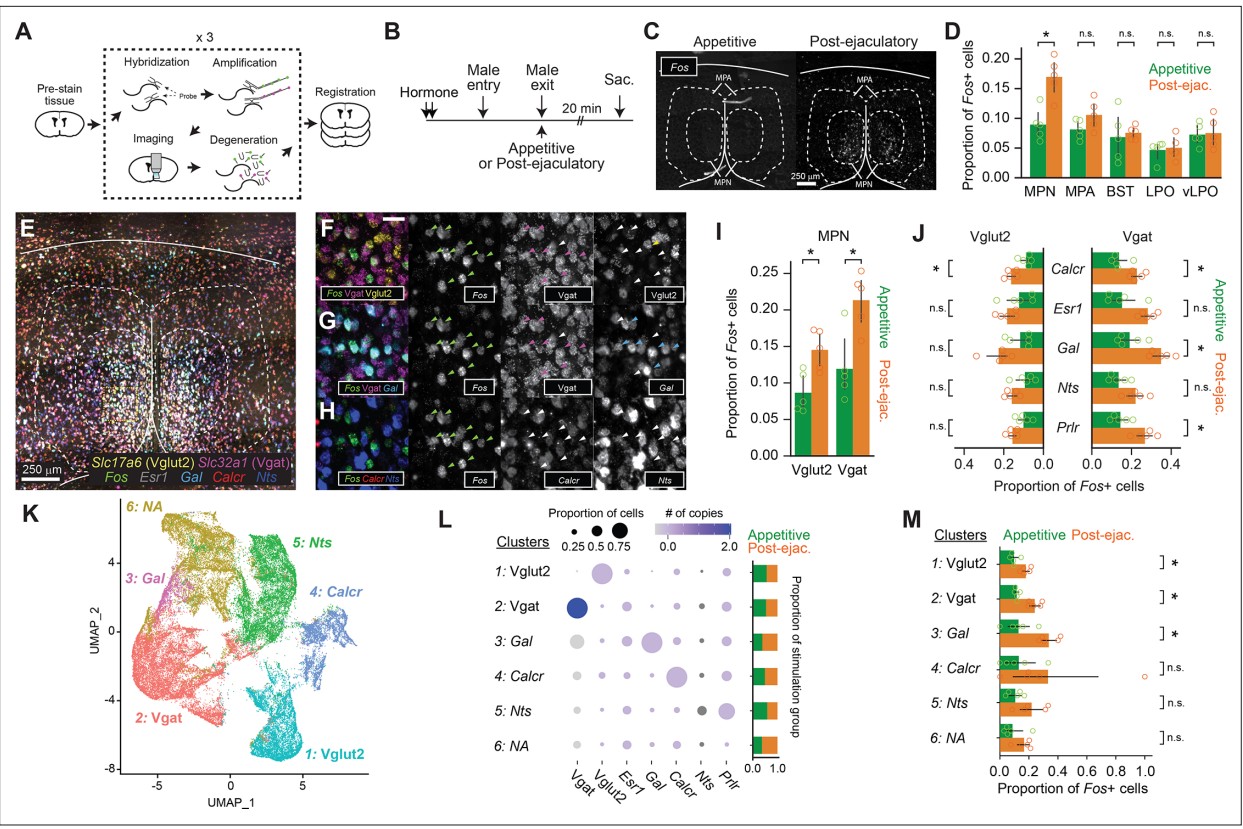

**Figure 3.** Male ejaculation responding cells are composed of a subset of excitatory and inhibitory cells in the medial preoptic area (MPOA). (**A**) Brain sections went through multiplexed *in situ* RNA hybridization chain reaction (HCR) to identify cell type markers of MPOA and an immediate early gene *Fos. Slc32a1* (Vgat), *Slc17a6* (Vglut2), *Esr1, Gal, Calcr, Nts, Prlr* were used as cell type markers. (**B**) Schematic of tissue sampling for *Fos* expression analysis in the MPOA. Wild-type female subjects were administered with hormones before the experiment day. On the experiment day, male partners entered the female home cage. Male partners were removed immediately after ejaculation ('post-ejaculatory' group) or during the appetitive phase before they showed mounting ('appetitive' group). Brain tissue was harvested 20 min after the male exit. (**C**) Representative coronal section showing *Fos* mRNA expression in the MPOA of Appetitive and Post-ejaculatory group. Scale bar, 250 µm. (**D**) Proportion of *Fos+* cells in each brain subregion. MPN: medial preoptic nucleus, MPA: medial preoptic area, BNST: bed nucleus of stria terminalis, LPO: lateral preoptic area, vLPO: ventral lateral preoptic area. BNST; t(8)=-0.369, p=1. LPO; t(8)=-0.3094, p=1. MPA; t(8)=-1.841, p=0.514. MPN; t(8)=-4.2709, *p=0.0136. VLPO; t(8)=-0.1751, p=1. (**E**) Representative coronal section showing expression of seven genes: *Fos,* Vgat, Vglut2*, Esr1, Gal, Calcr, Nts.* Scale bar, 250 µm. (**F–H**) Expanded image of area highlighted in yellow in (**E**). (**F**) Expression of *Fos,* Vgat, Vglut2. (**G**) Expression of *Fos,* Vgat and *Gal.* (**H**) Expression of *Fos, Calcr,* and *Nts.* Colored arrow indicates cells expressing given gene. Scale bar, 50 µm. (**I**) Proportion of Vgat or Vglut2 and *Fos+* cells in the MPN of Appetitive and post-ejaculatory group. Vglut2; t(8)=-3.3885, *p=0.019. Vgat; t(8)=-3.5361, *p=0.0153. (**J**) Proportion of Vgat+ *Fos* + or *Vglut2* + *Fos+* and cell type marker expressing cells in the MPN of Appetitive and post-ejaculatory group. Vgat + Calcr+ ; t(8)=-3.6646, *p=0.0318. Vgat +Esr1+; t(8)=-3.1442, p=0.0686. Vgat +Gal+ ; t(8)=-4.0896, *p=0.0174. Vgat +*Nts+* ; t(8)=-2.6382, p=0.149. Vgat +*Prlr*+ ; t(8)=-3.9487, *p=0.0212. Vglut2 +*Calcr*+ ; t(8)=-3.9487, *p=0.0146. Vglut2 +*Esr1*+; t(8)=-3.9487, p=0.8223. Vglut2 +*Gal*+ ; t(8)=-3.9487, p=0.1299. Vglut2 +*Nts*+ ; t(8)=-3.9487, p=0.1951. Vglut2 +*Prlr*+ ; t(8)=-3.9487, p=0.1038. (**K**) UMAP plot of MPN cells. (**L**) Disc plot showing proportion of gene expressing cell and the number of copies per cell for each gene and cluster. The proportion of cells from each group are shown on the right. Cluster *Vglut2*+; t(8)=-4.1631, *p=0.0189. Cluster Vgat+; t(8)=-5.9018, **p=0.0022. Cluster *Gal*+; t(8)=-3.9208, *p=0.0265. Cluster *Calcr*+; t(8)=-1.5438, p=0.9993. Cluster *Nts*+; t(8)=-2.1595, p=0.377. Cluster N.A.; t(8)=-1.8254, p=0.6322. (**M**) Proportion of *Fos* + cells from Appetitive and post-ejaculatory group in each cluster. All data are shown as mean ± 95% confidence interval and were analyzed by Student's t-test with Bonferroni correction (**D, E, K and N**). appetitive group, n=5. post-ejaculatory group, n=5. *p<0.05, **p<0.01, ***p<0.001; ns, not significant.

1: Vglut2, type 2: Vgat, and type 3: *Gal* had significantly more *Fos+* cells from the post-ejaculatory group compared to the appetitive group (**Figure 3M**). In contrast, *Fos+* cells were not induced in type 4: *Nts* indicating that MPOA neurons which regulate appetitive sexual behavior were not activated by male ejaculation. Collectively, these results indicate that male ejaculation induces more neural activity in the MPN when compared to appetitive behavior, both in excitatory and inhibitory cell types.

## A subpopulation of MPOA Vgat cells responds to male ejaculation in the female brain

While mapping neural activity based on *Fos* activity provided critical insight into how the brain responds to male ejaculation, the low temporal resolution of the method makes it difficult to identify cells that specifically respond of the onset to each behavior. For this reason, we next conducted single-cell resolution *in vivo* imaging in a freely moving animal (*Figure 4A and B*). We virally labeled MPOA inhibitory and excitatory cells with a calcium indicator GCaMP6s by using a Cre-dependent virus and female Vgat-Cre or Vglut2-Cre, respectively (*Figure 4C*, *Figure 4—figure supplement 1*) (Vgat-Cre: n=6, 87±27 cells. Vglut2-Cre: n=4, 68±42 cells). We recorded the calcium activity of MPOA Vgat and Vglut2 cells in female mice during mating in a home cage mating assay (*Figure 4E*). First, we analyzed the magnitude of calcium activity for 15 s after behavior onset. From peri-event analysis, we found that as a population, Vgat cells showed significant changes in calcium activity after the onset of all behaviors observed, with the highest increase after male ejaculation (*Figure 4F–G*). Vglut2 cells showed increased activity after sniffing and mounting, intromission, and male ejaculation. Furthermore, when we compared the response of Vgat and Vglut2 cells, Vgat cells showed significantly higher average response magnitude after sniffing, self-grooming, anogenital sniffing, and male ejaculation than the Vglut2 cells (*Figure 4H*). Vgat cells also contained a significantly higher proportion of cells that positively respond (>2$\delta$) to sniff and male ejaculation onset (*Figure 4I*). These results suggest that the male ejaculation signal is more dominantly represented in the MPOA inhibitory cells than in the excitatory cells. On the other hand, the response magnitude and the proportion of cells that respond after mounting and intromission had no significant difference between the two cell types. This indicates that the excitatory/inhibitory balance in the MPOA fluctuates during mating. The MPOA has high inhibitory activity during an appetitive phase, with more Vgat cells responding to sniffing, similar activity during the consummatory phase, and higher inhibitory activity after male ejaculation.

To further investigate the heterogeneity of MPOA during mating, we subset the cells into those which positively responded to any of sniff, mount, or male ejaculation, and then classified the cells based on their specificity (*Figure 5A, B*, *Figure 5—figure supplement 1*). Consistent with *Figure 4I*, the number of cells that specifically responded was largest to male ejaculation with 177 cells, next to mount with 46 cells, and to sniff with 12 cells. When compared to other cell populations, sniff cells and completion cells only showed significantly stronger responses than the other cells to sniff and male ejaculation, respectively (*Figure 5C*). Mount responsive cells showed significantly stronger response than the other cells to mount, intromission and anogenital sniffing, but not to sniffing and male ejaculation (*Figure 5C*). We also found that the cells that respond to these three behaviors partially overlap but were mostly distinct (*Figure 5D*). The proportion of cells that respond specifically to a behavior was highest for male ejaculation (Vgat: 80%, Vglut2: 82%), and lower in mounting (Vgat: 55%, Vglut2: 68%) and sniffing (Vgat: 36%, Vglut2: 0%). There was no notable bias in the anatomical distribution of these cells within the field of view (*Figure 5E*). Overall, these results suggest that the MPOA contains highly heterogenous functional population which contains a large proportion of cells specifically responding to male ejaculation in female mice.

## MPOA Vgat cells display prolonged activity after the onset of male ejaculation

Female mice avoid sexual interaction with a male animal after male ejaculation for hours and even days, suggesting a persistent changes in the animal's behavior (*McGill and Coughlin, 1970*; *Zhou et al., 2023*). How the MPOA regulates this persistent change in female sexual motivation is unclear. To analyze the response of MPOA cells on a longer time scale, we focused on the cells that positively responded to male ejaculation and characterized the calcium response pattern for 120 s after the onset (*Figure 6*). First, we observed that Vgat cells showed more diverse response patterns during this time-window (*Figure 6A and B*). To quantify this difference, we calculated the decay length of calcium signal, the time for the increased calcium signal to return to baseline (*Figure 6C*). As a result, we found that Vgat cells had significantly longer decay length compared to the Vglut2 cells, suggesting increased activity late after the onset of male ejaculation. To further classify the temporal dynamics of responses to male ejaculation, we first concatenated calcium activity that occurred within this time-window from both Vgat and Vglut2 cells and conducted spectral clustering in an unbiased fashion. As a result, we were able to classify the responses into five types that could be characterized as: (1)

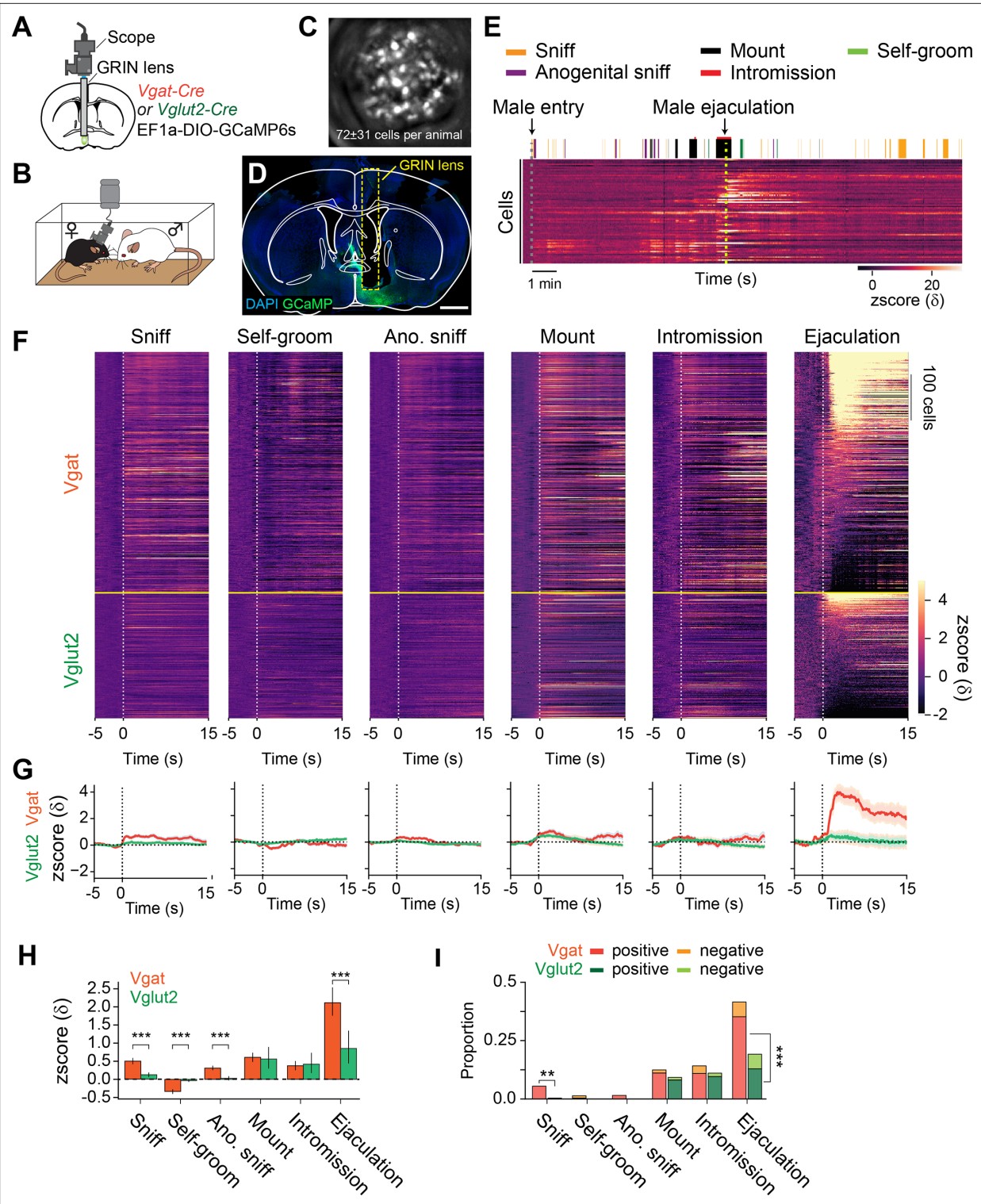

**Figure 4.** Male ejaculation signal is strongly represented in the MPOA of the female mice. (**A**) Schematic of *in vivo* calcium imaging from medial preoptic area (MPOA) Vgat and Vglut2 neurons. AAVDj EF1a-DIO-GCaMP6s was injected into the MPOA of Vgat-Cre or Vglut2-Cre female mice. A GRIN lens was placed above the MPOA following virus injection. (**B**) Calcium activity was imaged from a female subject during a free moving mating assay with a sexually experienced male partner. (**C**) Representative maximum projected image of the field of view during calcium imaging. Vgat: 87±28 cells, n=6. Vglut2: 68±42 cells, n=4. (**D**) Representative coronal section showing GCaMP6s expression and GRIN lens placement. (**E**) Representative raster plot showing behavior events during the mating assay and a heatmap showing calcium activity from cells imaged during the assay. Behaviors quantified: female-to-male sniffing (Sniff), female self-grooming (Self-groom), male-to-female anogenital sniffing (Ano. sniff), male

*Figure 4 continued on next page*

*Figure 4 continued*

mounting (Mount), male intromission (Intromission) and male ejaculation (Ejaculation). (**F**) Heatmap showing peri-event calcium activity around behavior onset. Cells are ordered by the average response after male ejaculation. (**G**) Line plot showing average calcium activity around behavior onset. Vgat neurons, blue. Vglut cells, orange. (**H**) Average calcium response magnitude following behavior onset per cell. Vgat-Cre: n=522 cells, Vglut2-Cre: n=271 cells. Sniff: t(791)=6.46, ***p=1.08E-09. Grooming: t(791)=-7.25, ***p=5.89E-12. Male anogenital sniffing: t(791)=6.64, ***p=3.41E-10. Mount: t(791)=0.34, p=1. Intromission: t(791)=-0.32, p=1. Ejaculation: t(791)=4.00, ***p=0.00042. (**I**) Proportion of cells which responded positively or negatively to behavior onset (>2δ). Positive: Sniff; Vgat 29/522 cells vs. Vglut2 1/271 cells, **p=0.00169. Grooming; Vgat 2/522 cells vs. Vglut2 0/271 cells, p=1. Male Anogenital Sniffing; Vgat 9/522 cells vs. Vglut2 0/271 cells, p=0.178. Mount; Vgat 58/522 cells vs. Vglut2 22/271 cells, p=1. Intromission; Vgat 57/522 cells vs. Vglut2 26/271 cells, p=1. Ejaculation; Vgat 184/522 cells vs. Vglut2 35/271 cells, ***p=1.52E-10. Negative: Sniff; Vgat 0/522 cells vs. Vglut2 0/271 cells, no statistic. Grooming; Vgat 5/522 cells vs. Vglut2 0/271 cells, p=0.636. Male Anogenital Sniffing; Vgat 0/522 cells vs. Vglut2 0/271 cells, no statistic. Mount; Vgat 7/522 cells vs. Vglut2 3/271 cells, p=1. Intromission; Vgat 17/522 cells vs. Vglut2 4/271 cells, p=0.831. Ejaculation; Vgat 33/522 cells vs. Vglut2 17/271 cells, p=1. All data are shown as mean ± 95% confidence interval and were analyzed by Student's t-test with Bonferroni correction (**H**) and chi-square test with Bonferroni correction (**I**). *p<0.05, **p<0.01, ***p<0.001; ns, not significant. Scale bar, 1 mm.

The online version of this article includes the following figure supplement(s) for figure 4:

**Figure supplement 1.** Lens placement and field of view of *in vivo* imaging experiment.

---

fast response, (2) slow response, (3) late response, (4) weak response, and (5) negative response (*Figure 6E*, *Figure 6—figure supplement 1*). Type 3 contained cells that showed increased calcium dynamics late after the onset of male ejaculation and had significantly larger decay length than the other cell types (*Figure 6F*). Then we separated the Vgat and Vglut2 cells to compare their proportions within each cluster. Interestingly, all the cells in type 3 were Vgat cells, suggesting that the increased activity in the MPOA late after the onset of male ejaculation is mediated by inhibitory neurons (*Figure 6H–K*). In contrast, Vglut2 cells were more likely to show fast responses than Vgat cells (*Figure 6K*). Collectively, these results suggest that the MPOA contains a subpopulation of inhibitory neurons that increase their activity late after the onset of male ejaculation, which may contribute to persistent suppression of the female animals' sexual motivation.

## An activity-defined subset of neurons in the MPOA is sufficient to suppress female sexual behavior

Next, we examined whether the increased activity of neurons in the MPOA is sufficient to suppress female sexual motivation after male ejaculation. We injected an AAV to express the chemogenetic neural activator hM3Dq or eYFP as control into the MPOA of TRAP2 female mice (*Figure 7A*). To capture the neural activity, each mouse was administrated 4-OHT immediately after the after male ejaculation ('post-ejaculatory-hM3Dq' or 'post-ejaculatory-eYFP' group). In addition to these animals, we also prepared a group of animals in which we injected an AAV expressing hM3Dq and then administrated 4-OHT during the appetitive phase ('appetitive-hM3Dq' group) or the consummatory phase ('consummatory-hM3Dq' group) (*Figure 7C*). We found that while the overall number of TRAP-labeled cells had no difference between the post-ejaculatory groups, consummatory group and appetitive group, the proportion of cells located in the MPN was significantly higher in the post-ejaculatory group (*Figure 7D*, *Figure 7—figure supplement 1A-C*). From *in situ* RNA hybridization, we found that most TRAP-labeled cells in the MPN were Vgat (Vgat: 0.76±0.16. Vglut2: 0.15±0.13). The proportion of Vgat cells was significantly higher than the Vglut2 cells (*Figure 5—figure supplement 1D–G*). Together these results suggest that the post-ejaculatory group labels more inhibitory neurons in the MPN than in appetitive group. Furthermore, we analyzed the accuracy and efficiency of TRAP labeling by allowing the female subject to experience male ejaculation before perfusion (*Figure 7—figure supplement 1H–L*). By immunolabeling c-Fos protein, we analyzed how well the TRAP ensemble overlaps with the c-Fos positive-ensemble in the MPN. As a result, we first found that the proportion of c-Fos + and TRAP-labeled cells in the post-ejaculatory group tended to be higher than the appetitive group and the consummatory group (*Figure 7—figure supplement 1I*). The proportion of c-Fos + cells were consistent across each group (*Figure 7—figure supplement 1J*). Next, we found that the accuracy of TRAP labeling (the proportion of TRAP-labeled cells that were also c-Fos positive) was similar between the groups (*Figure 7—figure supplement 1K*). The efficiency of TRAP labeling (the proportion of c-Fos + cells that were also TRAP labeled) tended to be higher in the post-ejaculatory group (*Figure 7—figure supplement 1L*) than in the appetitive group and the consummatory group. This also suggests that the appetitive group labels a different set of neurons than the post-ejaculatory

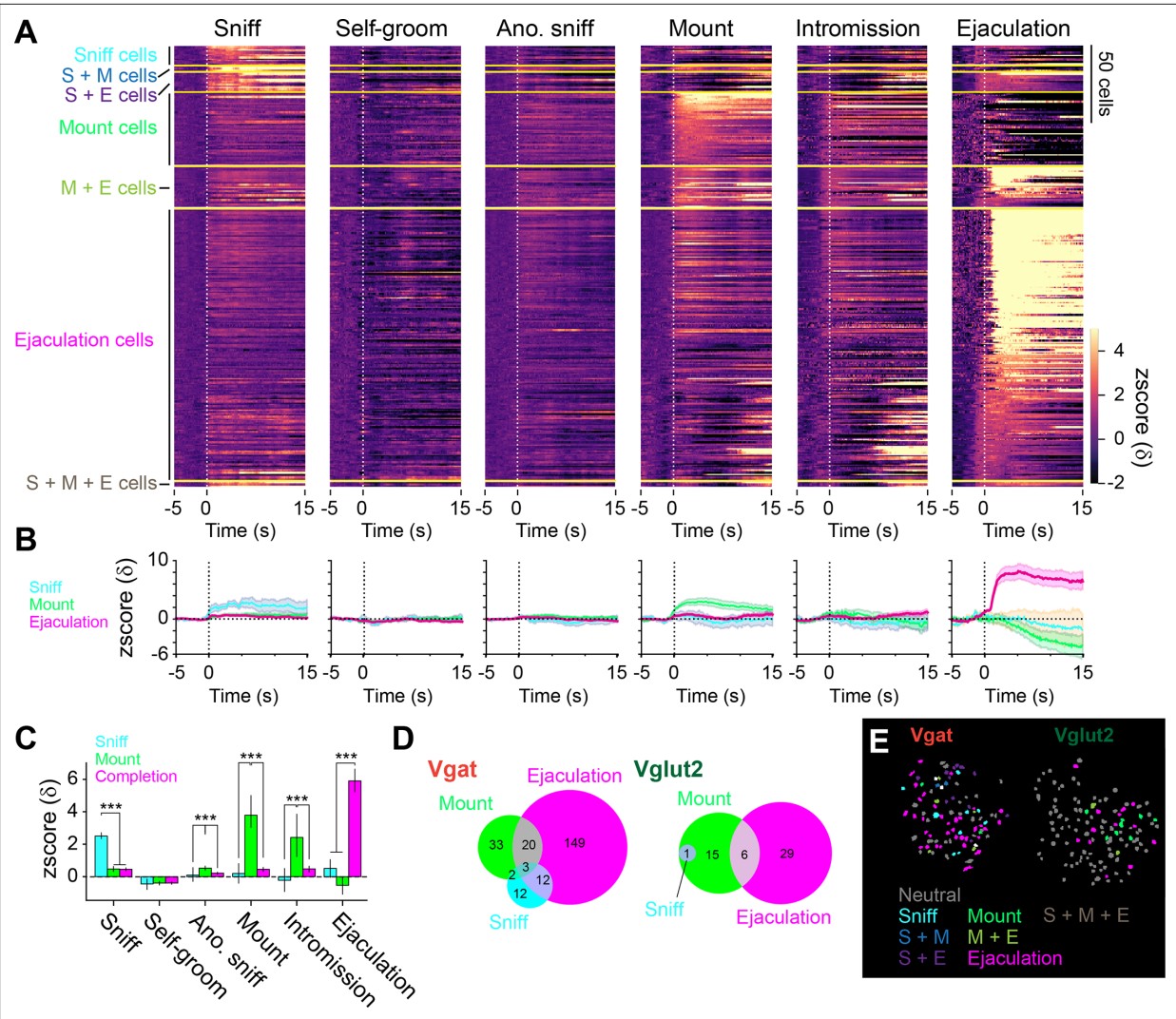

**Figure 5.** Heterogenous response properties to sexual behaviors in the medial preoptic area (MPOA). (**A**) Heatmap showing peri-event calcium activity around behavior onset of cells that were categorized based on response magnitude. S: Sniff, M: Mount, E: Ejaculation. (**B**) Line plot showing average calcium activity around behavior onset. Sniff cells, blue. Mount cells, green. Ejaculation cells, magenta. (**C**) Average calcium response magnitude following behavior onset per cell in each cell types. Sniff: One-way ANOVA, $F_{(2,235)}=73.35$, ***$p=1.77E-25$. Tukey's HSD test, Ejaculation cells vs Mount cells, $p=0.9996$. Ejaculation cells vs Sniff cells, ***$p=0$. Mount cells vs Sniff cells, ***$p=0$. Grooming: One-way ANOVA, $F_{(2,235)}=0.09$, $p=0.915$. Tukey's HSD test, Ejaculation cells vs Mount cells, $p=0.9723$. Ejaculation cells vs Sniff cells, $p=0.9236$. Mount cells vs Sniff cells, $p=0.9695$. Male Anogenital Sniffing: One-way ANOVA, $F_{(2,235)}=12.34$, ***$p=8.01E-06$. Tukey's HSD test, Ejaculation cells vs Mount cells, ***$p=0$. Ejaculation cells vs Sniff cells, $p=0.6794$. Mount cells vs Sniff cells, **$p=0.0048$. Mount: One-way ANOVA, $F_{(2,235)}=68.32$, ***$p=4.1E-24$. Tukey's HSD test, Ejaculation cells vs Mount cells, ***$p=0$. Ejaculation cells vs Sniff cells, $P=0.8726$. Mount cells vs Sniff cells, ***$p=0$. Intromission: One-way ANOVA, $F_{(2,235)}=15.72$, ***$p=3.9E-07$. Tukey's HSD test, Ejaculation cells vs Mount cells, ***$p=0$. Ejaculation cells vs Sniff cells, $p=0.5451$. Mount cells vs Sniff cells, ***$p=0.0009$. Ejaculation: One-way ANOVA, $F_{(2,235)}=49.59$, ***$p=1.08E-18$. Tukey's HSD test, Ejaculation cells vs Mount cells, ***$p=0$. Ejaculation cells vs Sniff cells, ***$p=0.0001$. Mount cells vs Sniff cells, $p=0.7295$. (**D**) Venn diagram showing the number of cells with positive and negative response in each cell type. ($>2\delta$). (**E**) Representative field of view showing spatial location of cells with different response properties to sexual behavior in Vgat (left) and Vglut2 (right) cells. All data are shown as mean ± 95% confidence interval and were analyzed by Student's t-test with Bonferroni correction (**C**). Sniff cells: n=12 cells. Mount cells: n=48 cells. Ejaculation cells: n=178 cells. *$p<0.05$, **$p<0.01$, ***$p<0.001$; ns, not significant.

The online version of this article includes the following figure supplement(s) for figure 5:

**Figure supplement 1.** Distribution of subjects per cluster.

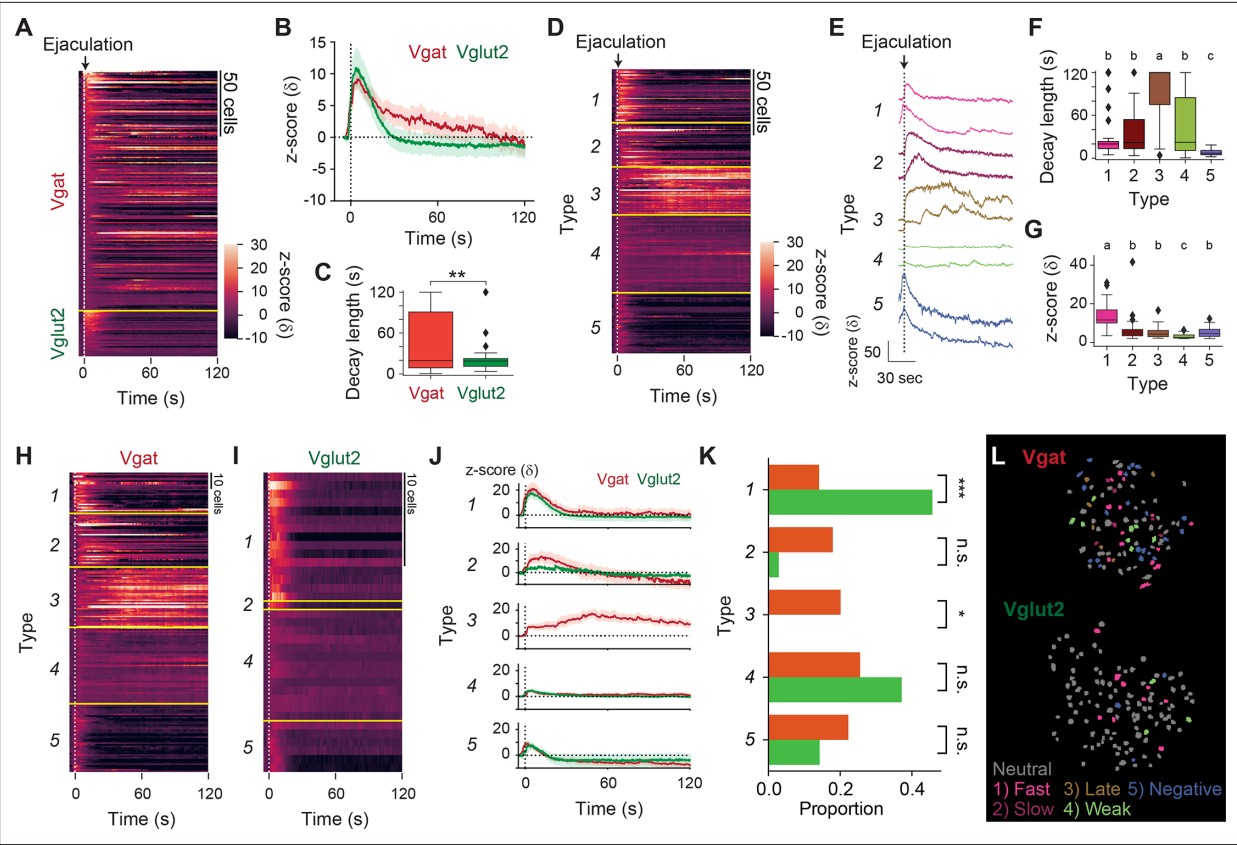

**Figure 6.** Medial preoptic area (MPOA) inhibitory neurons display prolonged activity late after the onset of male ejaculation in female mice. (**A**) Heatmap showing peri-event calcium activity around male ejaculation onset for male ejaculation-responding cells in a longer time scale. (**B**) Line plot showing average calcium activity of Vgat and Vglut2 population around behavior onset. (**C**) The decay length for male ejaculation-responding cells from Vgat and Vglut2 population. Vgat-Cre: n=184 cells. Vglut2-Cre: n=35 cells. t(217)=3.039653717, **p=0.002659548. (**D–G**) Cluster-based analysis of the activity pattern of the male ejaculation-cells. (**D**) Heatmap showing peri-event calcium activity around male ejaculation onset sorted by cluster. (**E**) Representative calcium activity traces of cells from each cluster. (**F**) The decay length for male ejaculation-responding cells from each cluster. Further details of statistical tests are shown in **Supplementary file 1**. (**G**) Average z-scored response for male ejaculation-responding cells from each cluster. Further details of statistical tests are shown in **Supplementary file 1**. (**H–K**) Cluster-based analysis of the activity pattern of the male ejaculation-cells in Vgat and Vglut2 population. (**H–I**) Heatmap showing peri-event calcium activity of Vgat (**H**) and Vglut2 (**I**) cells around male ejaculation onset sorted by cluster. (**J**) Line plot showing average calcium activity of Vgat and Vglut2 cells in each cluster around behavior onset. (**K**) Proportion of Vgat and Vglut2 cells in each cluster over Vgat and Vglut2 cells in all clusters. Statistical results are shown next to the cluster name. Type1; Vgat 26/184 cells vs Vglut2 16/35 cells, ***p=6.80E-05. Type2; Vgat 33/184 cells vs Vglut2 1/35 cells, p=0.119810823. Type3; Vgat 37/184 cells vs Vglut2 0/35 cells, *p=0.018. Type4; Vgat 47/184 cells vs Vglut2 13/35 cells, p=0.792188347. Type5; Vgat 41/184 cells vs Vglut2 5/35 cells, p=1. (**L**) Representative field of view showing spatial location of cells with different response properties after male ejaculation in Vgat (left) and Vglut2 (right) cells. All data are shown as mean ± 95% confidence interval and were analyzed by Student's t-test (**C**), ANOVA with post-hoc Tukey HSD test (**F and G**), and chi-square test with Bonferroni correction (**K**). Type 1: n=42 cells, Vgat-Cre: n=26 cells, Vglut2: n=16 cells. Type 2: n=34 cells, Vgat-Cre: n=33 cells, Vglut2: n=1 cells. Type 3: n=37 cells, Vgat-Cre: n=37 cells, Vglut2: n=0 cells. Type 4: n=60 cells, Vgat-Cre: n=47 cells, Vglut2: n=13 cells. Type 5: n=46 cells, Vgat-Cre: n=41 cells, Vglut2: n=5 cells. (**C and K**) *p<0.05, **p<0.01, ***p<0.001; ns, not significant. (**F and G**) a vs. b: p<0.001. b vs. c: p<0.05. a vs. c: p<0.001.

The online version of this article includes the following figure supplement(s) for figure 6:

**Figure supplement 1.** Distribution of subjects per cluster.

group. These results indicate that by using the TRAP2 method, we were able to specifically label a proportion of MPOA neurons, which are primarily inhibitory, that contains neurons that respond to male ejaculation.

To test the causality of these neurons, we first investigated how the activation affects female sexual behavior in a home cage mating assay (*Figure 7—figure supplement 2A–F*). The subjects were intraperitoneal injection (i.p.) injected with CNO 30 min before the assay. As a result, first we found that post-ejaculatory-hM3Dq group but not the appetitive-hM3Dq and consummatory-hM3Dq group showed significantly fewer sniffing events compared to post-ejaculatory-eYFP group

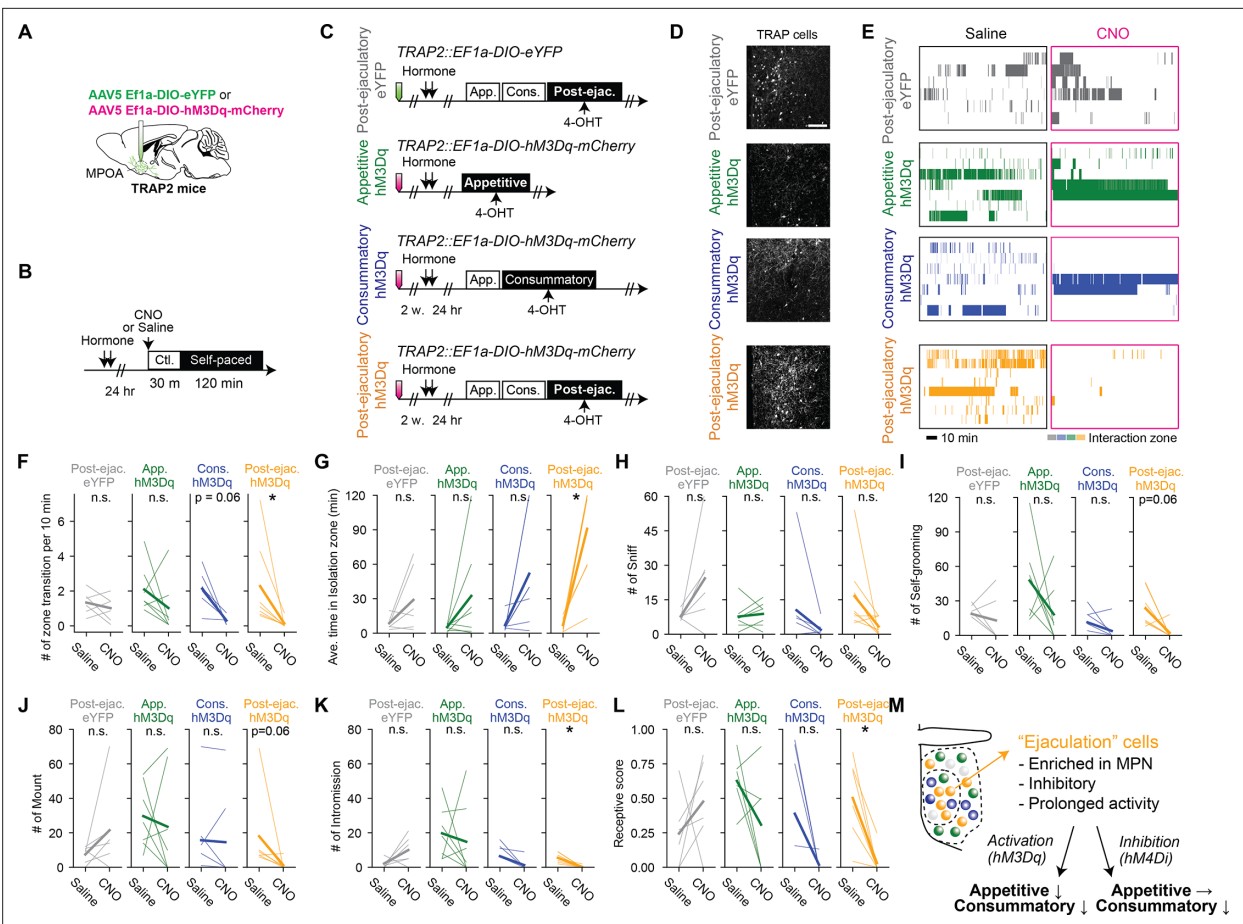

**Figure 7.** Activation of a subpopulation of medial preoptic area (MPOA) labeled following mating is sufficient to suppress female sexual behavior. (**A**) Schematic of selective labeling of MPOA neurons using TRAP2. AAV5 EF1a-DIO-hM3Dq-mCherry or AAV5 EF1a-DIO-eYFP as control was injected into the MPOA of the female TRAP2 subjects. After incubation of the virus, the female subjects went through procedures to targeted recombination in active populations method (TRAP) cells in MPOA (further described in **C**). (**B**) Schematic of pharmaco-genetic activation of MPOA neurons during a female self-paced mating assay. Female subjects were primed with hormone before the experiment day. On the experiment day, subjects were injected i.p. with clozapine-N-oxide (CNO) to activate TRAP cells, or saline as control. 30 min after the injection, the male partner was placed into the apparatus and the animal behavior was recorded for 2 hr. (**C**) After incubation of the virus, the female subjects were administered with hormones, then received an ejaculation from the male partner or only interacted without mating. Immediately after, the subjects were administrated with 4-hydroxytamoxifen (4-OHT). (**D**) Representative image showing TRAP cells MPN from post-ejaculatory-eYFP, appetitive-hM3Dq and post-ejaculatory-hM3Dq group. Further histological analysis is shown in **Figure 7—figure supplement 1**. Scale bar, 100 μm. (**E**) Raster plots of time spent in the interaction zone the female self-paced mating assay (post-ejaculatory-eYFP: gray. appetitive-hM3Dq: green, consummatory-hM3Dq: blue, post-ejaculatory-hM3Dq: orange). (**F**) Number of zone transitions per 10 min. Post-ejaculatory-hM3Dq; *p=0.03125. Post-ejaculatory-eYFP; p=1. Appetitive-hM3Dq; p=0.875. Consummatory-hM3Dq; p=0.0625. (**G**) Average time in isolation zone. Post-ejaculatory-hM3Dq; *p=0.03125. Post-ejaculatory-eYFP; p=1. Appetitive-hM3Dq; p=0.875. Consummatory-hM3Dq; p=0.125. (**H**) Number of female-to-male sniff. Post-ejaculatory-hM3Dq; p=0.11197. Post-ejaculatory-eYFP; p=0.375. Appetitive-hM3Dq; p=1. Consummatory-hM3Dq; p=0.4594. (**I**) Number of female self-grooming. Post-ejaculatory-hM3Dq; p=0.0625. Post-ejaculatory-eYFP; p=1. Appetitive-hM3Dq; p=1. Consummatory-hM3Dq; p=0.552. (**J**) Number of male mount. Post-ejaculatory-hM3Dq; p=0.0625. Post-ejaculatory-eYFP; p=0.875. Appetitive-hM3Dq; p=1. Consummatory-hM3Dq; p=1. (**K**) Number of male intromission. Post-ejaculatory-hM3Dq; *p=0.03125. Post-ejaculatory-eYFP; p=0.625. Appetitive-hM3Dq; p=1. Consummatory-hM3Dq; p=0.2716. (**L**) Receptive score (number of mount episodes with intromissions/number of all mount episodes). Post-ejaculatory-hM3Dq; p=0.03125. Post-ejaculatory-eYFP; p=1. Appetitive-hM3Dq; p=0.625. Consummatory-hM3Dq; p=0.2716. (**M**) Schematic illustration summarizing how MPOA neurons suppress female sexual motivation after male ejaculation. All data are shown by thin lines for individual subjects and thick lines for mean. The data were analyzed by Wilcoxon rank sum test with Bonferroni correction (**F–L**). post-ejaculatory-eYFP: n=6, appetitive-hM3Dq: n=7, consummatory-hM3Dq: n=8. post-ejaculatory-hM3Dq: n=8. *p<0.05, **p<0.01, ***p<0.001; ns, not significant.

The online version of this article includes the following figure supplement(s) for figure 7:

**Figure supplement 1.** Histological analysis of targeted recombination in active populations method (TRAP) labeling.

**Figure supplement 2.** MPOA[post-ejaculatory] cells supress female sexual behavior in home cage mating assay but not other behaviors.

**Figure supplement 3.** MPOA[post-ejaculatory] cells is not necessary for sexual motivation but for sexual receptivitiy.

(*Figure 7—figure supplement 2C*). For consummatory behaviors, while post-ejaculatory-hM3Dq group showed no difference in mounting episodes compared to post-ejaculatory-eYFP group, there were significantly less intromission episodes, resulting in lowerer receptive scores (*Figure 7—figure supplement 2C–F*). Thus, from these results, activation of MPOA neurons that respond to male ejaculation suppresses both appetitive and consummatory sexual behavior in female mice. As a next step, we investigated how the activation of these MPOA neurons affects female sexual motivation in the self-paced mating assay (*Figure 7*). The subjects were i.p. injected with CNO or saline 30 minutes before the assay (*Figure 7B*). As a result, only post-ejaculatory-hM3Dq group showed significantly fewer number of transitions and spent more time in the isolation zone on the CNO trial than on the saline trial (*Figure 7E–G*). We also found that post-ejaculatory-hM3Dq group displayed fewer numbers of self-grooming, mounting, intromission and lower receptivity score on the CNO trial than on the saline trial (*Figure 7H–L*). Interestingly, activation of these neurons did not induce conditional place preference or aversion (*Figure 7—figure supplement 2G–J*), implying that the suppression of sexual behavior is not associated with negative valence. Next, we investigated how inhibition of these neurons affects female sexual behavior (*Figure 7—figure supplement 3*). We prepared a new group of mice in which we injected an AAV expressing hM4Di or eYFP as control in TRAP2 female mice. Two-weeks after injection, we administrated 4-OHT immediately after the subject experienced male ejaculation ('post-ejaculatory-hM4Di' or 'post-ejaculatory-eYFP' group). First, we tested the animals in the self-paced mating assay (*Figure 7—figure supplement 3A–L*). The female subjects were i.p. injected with CNO or saline 30 min before the assay (*Figure 7—figure supplement 3B*). As a result, the subjects did not show any significant change in sexual behavior during this assay (*Figure 7—figure supplement 3D–L*). Next, we tested the animals in the home cage mating assay (*Figure 7—figure supplement 3M–S*). The subjects were intraperitoneal injection (i.p.) injected with CNO 30 min before the assay (*Figure 7—figure supplement 3M*). As a result, the post-ejaculatory-hM4Di group experienced did not show significant changes in sniffing, self-grooming, or mounting, but showed more intromission and higher sexual receptivity than in the post-ejaculatory-eYFP group (*Figure 7—figure supplement 3O–S*). Collectively, these result suggests that the MPOA neurons are sufficient but not necessary for the modulation for sexual motivation but is sufficient and necessary for consummatory behaviors in female mice (*Figure 7M*).

## Discussion

How sexual motivation is represented in the brain has been a long-lasting question in the field. Here, we found that MPOA plays a critical role in regulating female sexual motivation. We found that MPOA neurons, especially the inhibitory neurons, respond significantly and specifically to male ejaculation. These neurons show persistent activity after the onset of male ejaculation. Reactivation of these neurons result in suppression of female sexual motivation. We propose that MPOA together with other neural substrates such as the BNST (*Zhou et al., 2023*), encode a negative feedback signal that sustains low sexual motivation state after male ejaculation in female mice. Here, we will further discuss how sensory inputs contribute to the male ejaculation response in the MPOA, how MPOA responds to different sexual behaviors and how these neurons work in concert with other sexual behavior-related hypothalamic regions.

### Using self-paced mating paradigm to evaluate sexual motivation in female rodents

The self-paced mating paradigm is a classical yet powerful method to directly assess female sexual behaviors (*Erskine, 1989*; *Erskine, 1985*; *Peirce and Nuttall, 1961*). This paradigm is still in current use, mainly in rats (*Ventura-Aquino and Paredes, 2023*). Our paper demonstrate that this simple behavioral paradigm is still a powerful method to study neural circuitry of female sexual behavior and motivation in mice. In our results, we first demonstrate that the latency to return to the interaction zone was significantly larger after the animal experienced male ejaculation consistent with the results shown in rats. Direct measurements of both increase of withdrawal and decrease of approach to sexual behavior are the strength of this paradigm. Furthermore, we have also demonstrated that female animals show more time spent in the isolation zone regardless of the male partners sexual motivation state (*Figure 1—figure supplement 2*). From these results, we believe that this paradigm

reports important metrics that correlate with female sexual motivation. However, it is also clear that other factors affect the females behavior in the self-paced mating paradigm. Recent studies developed an operant conditioning apparatus which delivers an intruder to a male subject in response to a nose poke (*Minakuchi et al., 2024*). This apparatus is a powerful approach to dissociate motivation and action which is difficult in commonly used self-paced paradigms. The self-paced mating paradigm used in our study has limitations to studying sexual motivation in detail. An advancement in the design of the apparatus may be essential to further study the neural circuitry that regulates female sexual motivation in fine detail.

## Sensory representation of male ejaculation in the female brain and genitalia

Suppression of female sexual behavior and motivation after male ejaculation is not a unique phenomenon found in mice and can be observed in various insect species (*Gillott, 2003*). Studies in *Drosophila* has demonstrated that the primary trigger of this behavioral switch is the sex peptide (SP), a peptide released in the male seminal fluid (*Chapman et al., 2003*; *Chen et al., 1988*; *Liu and Kubli, 2003*). Studies in flies have further identified the receptor for SP as well as the neural circuitry which conveys the sensory information to the mating regulating neurons in the brain (*Feng et al., 2014*; *Rezával et al., 2012*; *Yapici et al., 2008*). In contrast to the well-studied system in *Drosophila*, little is known about how the mammalian brain senses male ejaculation. Histological studies in rats have shown that male ejaculation induces c-Fos expression widely in the brain, including the MPOA (*Pfaus et al., 1993*). Indeed, anesthesia to the female genitalia has been shown to disrupt the pacing of mating, indicating that sensory input from the genitalia can impact female sexual behavior (*Parada et al., 2014*). Furthermore, artificial stimulation of the vagina or the clitoris of the female rat was further shown to induce c-Fos expression in the MPOA in a similar fashion as male ejaculation, indicating that at least tactile sensory input can be conveyed to this brain area (*Parada et al., 2010*; *Pfaus et al., 1996*). Ejaculation from vasectomized male animals were sufficient to suppress sexual behavior suggesting that at least chemical components from the testis is not necessary for this effect (*Zhou et al., 2023*). A recent study has shown that mechanical stretch induces serotonergic activity in the dorsal raphe, which resembles changes after male ejaculation, suggesting that stretch sensation is mediating male ejaculation signal to the brain (*Troconis et al., 2023*). Mouse genetics has also enabled identification of nerves innervating the genitalia (*Qi et al., 2024*). However, there are still to uncover on how stretch sensation in the genitalia is conveyed to the brain, and equally unclear how this information is ultimately transmitted to the MPOA. Our findings on how the male ejaculation signal is represented in the MPOA will be an ideal entry point to study the neural basis of genital-brain communication.

## Appetitive behavior and male ejaculation ensembles in MPOA

The MPOA is a molecularly and functionally heterogenous brain region. One function MPOA has heavily been related to is parenting behavior. MPOA neurons which express *Gal* and/or *Calcr* has been demonstrated to regulate parenting behavior (*Kohl et al., 2018*; *Wu et al., 2014*; *Yoshihara et al., 2021*). From our data, neurons that respond to male ejaculation tended to co-express *Gal* and *Calcr* more than neurons that responded to appetitive behavior (*Figure 2—figure supplement 1*). However, we also found that chemogenetic activation of male ejaculation responsive MPOA neurons did not induce parenting behavior (*Figure 7—figure supplement 2K*). While it is necessary to conduct extensive analysis of how MPOA neurons respond to other behaviors, these results suggest that male ejaculation responding MPOA neurons may specifically regulate sexual behavior and motivation but not parenting behavior. A previous study also found that MPOA contains a subset of neurons which regulate female appetitive sexual behavior. MPOA neurons which express *Slc32a1* (Vgat), *Esr1* and *Nts*, and projects to the ventral tegmental area (VTA) has been shown to respond to olfactory signals from the male mouse and to enhance female-to-male sniffing (*McHenry et al., 2017*). There is also a significant release of dopamine after the first female-to-male sniffing interaction as well as after male ejaculation (*Dai et al., 2022*). How the brain, especially the MPOA, increases sexual motivation before and during the appetitive phase, and how it suppresses motivation after male ejaculation, has been a long-lasting question in the field that is addressed throughout this work. The whole brain activity-dependent labeling experiment identified the MPOA as one of the brain regions which strongly increases Fos activity (*Figure 2*).

However, there were many other brain regions showing increased activity associated with male ejaculation, including ones that are not canonically involved in the regulation of social behavior. While it is possible that all these brain regions work in concert to suppress sexual motivation after male ejaculation, our results demonstrate that a specific brain region, MPOA, is sufficient to evoke behavioral response. This suggests that all brain regions showing higher Fos activity are not necessarily causal to behavioral changes. It is also possible that these widespread brain regions may be a consequence of noise due to the long time-window in this method, which may be resolved by further development of genetic methods for neural ensemble labeling. In the MPOA, our *in vivo* calcium imaging data shows nearly half of the inhibitory neurons that positively respond to appetitive behavior also respond to male ejaculation (*Figure 5*). Our data also identified that there was no difference between the number of *Nts+* cells activated by appetitive behavior and male ejaculation (*Figure 3*). These results suggest that there is a small but significant number of neurons which respond both to appetitive behavior and male ejaculation in the MPOA, and potentially regulate release of dopamine.

Although a large number of neurons that responded to female-to-male sniffing also respond to male ejaculation, the opposite is not true (*Figure 5*). Importantly, the chemogenetic activation of neurons that were active after male ejaculation suppressed the amount of sexual interaction in the self-paced mating assay, opposite to what was described when activating Nts+ MPOA neurons (*McHenry et al., 2017*). This suggests that there are at least two-distinct populations in the MPOA which regulate appetitive sexual behavior or sexual motivation in opposing directions.

## Neural circuitry for female sexual behavior regulation

The neural circuitry that regulates female sexual behavior has been extensively studied in rodent models. From early lesion and electrical stimulation studies, it has been proposed that the rodent brain has a positive and negative regulatory system for female sexual behavior. The positive system has been proposed in rats, where Pfaff and Sakuma demonstrated that the lesion of the ventromedial hypothalamus (VMH) abolished sexual receptivity while electrical stimulation of the same region induces receptivity (*Pfaff and Sakuma, 1979a*; *Pfaff and Sakuma, 1979b*). Mouse genetics further showed that a subset of neurons in the ventrolateral part of the VMH is crucial for the regulation of female sexual behavior (*Inoue et al., 2019*; *Yang et al., 2013*). More specifically, *Esr1+ Cckar+* (*Nts*) cells in the VMHvl were found to be necessary and sufficient for female sexual receptivity (*Liu et al., 2022*; *Yin et al., 2022*). These results suggest that the VMHvl is the 'command center' which positively regulate female sexual behavior.

The MPOA together with other brain regions has been proposed to form the negative regulating system. Lesion of the MPOA was shown to increase the sexual receptivity of the female rat and electrical stimulation was shown to reduce receptivity, suggesting that MPOA negatively regulates female sexual behavior (*Pfaff and Sakuma, 1979b*; *Powers and Valenstein, 1972*). In the current study, we demonstrate that chemogenetic activation of neurons in the MPOA which respond to male ejaculation is sufficient to suppress female sexual behavior and sexual motivation. This result suggests that these neurons in the MPOA constitute the negative regulating system. Importantly, both our single-cell calcium imaging results and histological analysis of the chemogenetic activation cohort suggest that the majority of these neurons are inhibitory (*Figure 4I* and *Figure 7—figure supplement 1G*). We further demonstrate that the inhibitory population contains neurons that show prolonged responses to male ejaculation which resemble the persistent change of sexual motivation. Other studies have also supported the idea that the inhibitory cell population is more related to the regulation of social behavior than the excitatory population (*Fang et al., 2018*; *Hashikawa et al., 2021*; *Moffitt et al., 2018*; *Tsuneoka and Funato, 2021*; *Zhang et al., 2021b*). Our results are also in line with this idea and suggest that the inhibitory neuronal population is the main regulator of the change in sexual motivation. The role of excitatory neurons that respond to male ejaculation will be a future topic to examine.

Recently, BNST was also found to play a role in negatively controlling female sexual behavior (*Zhou et al., 2023*). There, they use a behavioral assay which introduces 2 different males to the female subject before and after male ejaculation. They demonstrate that after male ejaculation, the female will be less receptive toward the second male. Importantly, this assay clearly demonstrate that the reduction of female sexual behavior is due to the female herself, not due to the change in their male conspecific. This result together with the result from our self-paced mating assay (*Figure 1*) strongly

indicates that the behavioral change after male ejaculation reflects change in internal motivational states. The study further used single-cell resolution *in vivo* calcium imaging techniques to demonstrate that BNST[Esr2] neurons show specific and robust responses after male ejaculation, like what we observed in the MPOA. They then use opto- and chemogenetic tools to study the causality of these neurons. They demonstrate that the inhibition of BNST[Esr2] neurons rescued female receptivity after male ejaculation, activation of these neurons did not affect sexual behavior in female mice, suggesting that BNST[Esr2] neurons are necessary but not sufficient to suppress female sexual behavior and sexual motivation. While there were some differences in how modulation of BNST and MPOA neurons affect sexual behavior and motivation, overall, the role of BNST and MPOA seems to be similar suggesting that the two regions work in concert to suppress female sexual behavior. This is contrary to the finding from Mei et al. showing that the BNST[Esr1] neurons and MPOA[Esr1] neurons forms a bidirectional inhibitory circuit to antagonistically regulate infanticidal behavior and maternal behavior, respectively (*Mei et al., 2023*). A mechanism which is yet to be revealed can override the negative feedback loop between MPOA and BNST neurons, potentially a disinhibitory circuit that connects the BNST[Esr2] and MPOA[GABA], is necessary for these neurons to regulate the same behavior change.

Interestingly, a subpopulation in the VMHvl expressing Cckar shows strong inhibition of spontaneous bulk calcium events after male ejaculation (*Yin et al., 2022*). This is opposite to the persistent increase of activity in the MPOA and BNST, suggesting that the activity in these regions is reflected onto the VMHvl. Indeed, MPOA and BNST both send dense projection to the VMHvl, and VMHvl[Esr1] neurons project to the MPOA and to the BNST, making the MPOA-BNST-VMHvl network highly interconnected (*Dimén et al., 2021*; *Lo et al., 2019*; *Osakada et al., 2018*). A recent study focusing on male aggression motivation further demonstrated that the long-range input from inhibitory MPOA neurons to VMHvl promote and prolongs low motivational state (*Minakuchi et al., 2024*). From this finding and the fact that the acute activation of MPOA neurons more drastically affected female sexual behavior than the BNST[Esr2] neurons, we speculate that the MPOA can directly inhibit the VMHvl, while the BNST utilizes a different circuit mechanism which either involves multiple synaptic connections or different molecular mechanisms to act on the VMHvl.

Long-lasting change in neural activity within this network, potentially through monoamine or steroid molecules such as dopamine or opioids may be the neural basis of suppressed sexual motivation after male ejaculation (*Micevych and Meisel, 2017*; *Zhang et al., 2021a*). Another candidate is serotonin, which is known to be an important mediator for sexual behavior (*Uphouse, 2014*). Recent study has revealed the release of serotonin in response to male ejaculation in female mice (*Troconis et al., 2023*). Our whole brain activity mapping data also indicates increased Fos activity in the dorsal raphe (*Figure 2—figure supplement 1* and *Figure 2—figure supplement 2*). Whether serotonin mediates the changes in persistent activity across the entire network and whether there is a difference in how it impacts MPOA and BNST and its output to the VMHvl will be an important topic of future studies.

Taken together, our study identifies a subset of neurons in the MPOA which is dedicated to negatively sustain female sexual motivation. This finding fills the missing hole in how the brain neurons regulate female sexual motivation and proposes that the basal forebrain complex of MPOA and BNST work in concert as the negative regulating system of female sexual behavior.

# Materials and methods

**Key resources table**

| Reagent type (species) or resource | Designation | Source or reference | Identifiers | Additional information |
|---|---|---|---|---|
| Strain, strain background (*Mus musculus*) | *Mus musculus* with name C57BL/6 J | https://www.jax.org/strain/000664 | RRID:IMSR_JAX:000664 | |
| Strain, strain background (*Mus musculus*) | *Mus musculus* with name Slc32a1tm2(cre)Lowl (vgat-cre) | https://www.jax.org/strain/016962 | RRID:IMSR_JAX:016962 | |
| Strain, strain background (*Mus musculus*) | *Mus musculus* with name Slc17a6tm2(cre)Lowl (vglut2-cre) | https://www.jax.org/strain/016963 | RRID:IMSR_JAX:016963 | |
| Strain, strain background (*Mus musculus*) | *Mus musculus* with name Fostm2.1(icre/ERT2) Luo/J (Fos2a-cre or TRAP2) | https://www.jax.org/strain/030323 | RRID:IMSR_JAX:030323 | |

*Continued on next page*

*Continued*

| Reagent type (species) or resource | Designation | Source or reference | Identifiers | Additional information |
|---|---|---|---|---|
| Strain, strain background (AAV5) | AAV5 hSyn-DIO-hM4D(Gi)-mCherry | https://www.addgene.org/44362/ | Catalog #: 44362-AAV5 | lot #: v178820 |
| Strain, strain background (AAV5) | AAV5 hSyn-DIO-hM3D(Gq)-mCherry | https://www.addgene.org/44361/ | Catalog #: 44361-AAV5 | lot #: v141469 |
| Strain, strain background (AAV5) | AAV5-Ef1a-DIO-eYFP | UNC Vector Core | | lot #: av4802B |
| Strain, strain background (AAVDj) | AAVDj EF1a-DIO-GCaMP6s | UNC Vector Core | | lot#: av78310 |
| Software, algorithm | Python 3.7 (Anaconda Distribution) | https://www.anaconda.com/ | | |
| Software, algorithm | R 4.0.4 | https://cran.r-project.org/ | | |
| Software, algorithm | Seurat3.1.1 | https://github.com/satijalab/seurat/releases/tag/v3.1.1; *Hao et al., 2024*; *satijalab, 2019* | | |
| Software, algorithm | DeepLabCut 2.2 | https://github.com/DeepLabCut/DeepLabCut; *Nath et al., 2019*; *Lauer et al., 2025* | | |
| Software, algorithm | SimBA 1.31 | https://github.com/sgoldenlab/simba; *Goodwin et al., 2024*; *Nilsson and sgoldenlab, 2025* | | |
| Chemical compound, drug | Easyindex | LifeCanvas Technologies | | |
| Chemical compound, drug | SHIELD Kit | LifeCanvas Technologies | | |
| Chemical compound, drug | Delipidation Buffer | LifeCanvas Technologies | | |
| Chemical compound, drug | Conduction Buffer | LifeCanvas Technologies | | |

## Animals

All experiments were approved by the Institutional Animals Care and Use Committee at the University of Washington (Protocol #4450–01). Animals were group-housed with littermates on a 12 hr light cycle at ~22° C with food and water available *ad libitum*. Wild-type Swiss-Webster mice were purchased from Taconic Biosciences (NY, U.S.). Wild-type C57BL/6 mice were purchased from the Jackson laboratory and bred in-house. Vgat-Cre (also known as *Slc32a1*-ires-Cre, Jax#016962), Vglut2-Cre (also known as *Slc17a6*-ires-Cre, Jax#028863) and Ai14 (*Rosa26*-CAG-LSL-tdTomato-WPRE, Jax#007914) were purchased from Jackson laboratory. TRAP2 mice (also known as Fos-2A-iCreER[T2], *DeNardo et al., 2019*) were provided by Richard Palmiter (University of Washington) and crossed with Ai14 to generate TRAP2::Ai14 mice. All male mice which were used as intruders were singly housed to avoid male-male aggression. Female mice with GRIN-lens implant were singly housed. All other mice were group-housed.

## Surgery

For targeting AAV into a certain brain region, stereotactic coordinates were defined for each brain region based on the brain atlas (*Franklin and Paxinos, 2007*). Mice were anesthetized isoflurane and head-fixed to stereotactic equipment (Kopf). For *in vivo* calcium imaging experiments, 500 nL of AAVDj EF1a-DIO-GCaMP6s (2.8*e12, UNC viral core, lot#av78310) was injected into two coordinates in the target brain region at a speed of 60 nl/min using Nanoject 3 pump (Drummond Scientific). The following coordinates were used [Distance in millimeters from the Bregma for the anterior (A)-posterior (P) and lateral (L) positions, and from the brain surface for ventral (V) direction]: MPOA, A 0.2, L 0.15, V 5.2. and A 0.4, L 0.35, V 5.2. Immediately following viral injection, five screws were placed into the skull and a GRIN lens (Inscopix, 0.6 mm × 7.3 mm) was slowly lowered to position above

the injection site: MPOA, A 0.3, L 0.3, V 4.95. GRIN lens was glued to the skull with C&B Metabond Kit (Parkell, # S380) and dental cement (Lang Dental, #1304 CL). After placement of GRIN lens, a custom-made head ring was glued to the skull and screws. For bare GRIN lenses, more than 3 wk after the virus injection and GRIN lens placement surgery, a baseplate (Inscopix) was positioned to while visualizing GCaMP6s expression with a microscope. Once a field of view with GCaMP6s expression was found, the baseplate was glued to the head ring and skull. For pharmacogenetic experiments, 300–400 nL of AAV5 hSyn-DIO-hM3D(Gq)-mCherry (2.0*e13, Addgene, #44361-AAV5, lot#v141469), AAV5 hSyn-DIO-hM4D(Gi)-mCherry (2.5*e13, Addgene, #44362-AAV5, lot#v178820), or AAV5 EF1a-DIO-eYFP (4.4*e12, UNC viral core, lot#av4802B) was bilaterally injected into the target brain region at a speed of 60 nl/min using Nanoject 3 pump. The following coordinates were used: MPOA, A 0.3, L 0.25, V 5.2. To avoid pregnancy during the first experience, all female mice that experienced a mating assay except ones used in *Figure 1N–T* was ovariectomized. Ovariectomized female mice that were primed with hormones has been successfully used in other sexual behavior studies (*Ring, 1944*; *Yang et al., 2013*). Animals that were used *in vivo* calcium imaging experiments or pharmacogenetic experiments were ovariectomized during the virus injection or GRIN lens placement surgery. After surgery, the incision was sutured, and the animal was warmed using a heat pad to facilitate recovery from anesthesia.

## Behavior experiments and analysis

For all mating assays, female mice older than 6 wk were ovariectomized (OVX) under isoflurane anesthesia. If necessary, mice underwent surgeries for AAV injections, GRIN lens implantation on the same day. After more than 2 wk of recovery, the OVX female subjects were subcutaneously (s.c.) injected with 0.1 mL and 0.05 mL Estradiol (E8515, Sigma-Aldrich, 0.1 mg/ml) dissolved in sesame oil (Sigma-Aldrich) at 24 and 48 hr before assay, and 0.05 mL Progesterone (P0130, Sigma-Aldrich, 10 mg/ml in sesame oil) at 4 hr before assay. If necessary, mice underwent I.P. injection of CNO/saline. C57BL/6 or Swiss-Webster mice were used as a stud male. A 4–6 cm to head bar was placed onto the male mice under isoflurane anesthesia, which was used for the female self-paced mating assay described below. The male animals were kept isolated until the assay and had been trained prior to the assay to show mounting and ejaculation.

The home cage mating assay was conducted as previously described with modifications (*Ishii et al., 2017*). Each assay was conducted 5–10 hr after the onset of dark period, under an IR light and recorded (30 Hz) for subsequent 30–120 min for analysis of sexual behavior from a dorsal view. For neural activation experiments using hM3Dq (*Figure 7—figure supplement 1*), 0.1 mL of 1 mg/kg clozapine-N-oxide dissolved in saline (CNO, RTI International, NIMH Code C-929) or 0.1 mL of saline was I.P. injected into the animal 30 min prior to the male entry. For neural inhibition experiments using hM4Di (*Figure 7—figure supplement 3*), 0.1 mL of 5 mg/kg CNO dissolved in saline or 0.1 mL of saline was I.P. injected into the animal 30 min prior to the male entry.

The female self-paced mating assay was conducted using a behavior apparatus modified from an apparatus used in rat (*Nyuyki et al., 2011*). A 1 cm thick black wall with a 20 mm × 20 mm hole 3D-printed with PFA filament was placed in a 25 cm × 35 cm cage to divide the room in a 1:2 ratio along the longer edge. The smaller area was defined as the isolation zone and the larger area was defined as the interaction zone. The female subjects that were not able to pass through the hole during the habituation was eliminated from the cohort. On the experiment day, the female subjects were placed into the behavior apparatus for 30 min for analysis of movement tracking from a dorsal view, which was used as control (Control trial). After the habituation, the partner (Swiss-Webster) was placed in the interaction zone. Each assay was conducted 5–10 hr after the onset of dark period, under an IR light and recorded (15 Hz) for subsequent 30–120 min for analysis of movement tracking and sexual behavior from a dorsal view (Sexual interaction trial). For *Figure 7*, females underwent two mating assays at >1 wk interval, with different treatment (e.g. saline or CNO). Different males were used for each assay. For neural activation experiments using hM3Dq (*Figure 7*, *Figure 7—figure supplement 1*, *Figure 7—figure supplement 2*), 0.1 mL of 1 mg/kg CNO dissolved in saline or 0.1 mL of saline was I.P. injected into the animal 30 min prior to the male entry. For neural inhibition experiments using hM4Di (*Figure 7—figure supplement 3*), 0.1 mL of 5 mg/kg CNO dissolved in saline or 0.1 mL of saline was I.P. injected into the animal 30 min prior to the male entry.

Sexual behavior was annotated from the recorded video using Behavioral Observation Research Interactive Software (BORIS) in a frame-by-frame fashion (*Friard and Gamba, 2016*). Male mounts, intromission, male-to-female anogenital sniffing, female-to-male sniffing, female self-grooming and male ejaculation were annotated based on characteristic postures described previously (*Ishii et al., 2017*). Briefly, mount was defined as a stud male using both forepaws to climb onto a female from behind for copulation. Intromission was defined by stable thrusting from the male animal during mount. Male ejaculation was identified by increase and then termination of thrusting and more than 10 s of immobility from the male, which are postures frequently associated with male ejaculation. Male-to- female anogenital sniffing was identified by the male animal following the female subjects anogenital area. Female-to-male sniffing was identified by female subject nose approaching male subject. When analyzing sniff-evoked GCaMP activity, to restrict the analysis to appetitive sniffing behaviors, sniffing behaviors which happened after the first mounting were excluded from the analysis. Female self-grooming was identified by the female subject moving their head close to their genital region. The receptive ratio was calculated by dividing the total number of intromissions with the total number of mountings. Behavior annotation was conducted by volunteers who were blind for the stimulation or AAV condition. For the female self-paced mating assay, the position of the female subject was tracked from the recorded video using DeepLabCut (*Nath et al., 2019*) or Ethovision. The amount of time spent in the interaction zone was quantified using SimBA region of interest (ROI) interface (*Goodwin et al., 2024*). Behavior data from *in vivo* calcium imaging experiments were downsampled to 10 Hz to match the imaging data.

For conditional place preference (*Figure 7—figure supplement 2G–J*), a two-chambered apparatus was used with visual (horizontal and vertical stripes) cues. The floor was backlit and an overhead video camera recorded position (Basler, 3.75 Hz frame rate with Ethovision software, Noldus). The apparatus was in a sound isolation chamber. *Ad libitum* fed Fos-CreER mice with hM3Dq and eYFP controls were acclimatized (60 min). The following day, the mouse's position was recorded (1800s) and tracked offline using Ethovision (pre-test). Daily conditioning consisted of two 1800s sessions: (1) one side of the chamber paired with saline injection; (2) the other side paired with CNO (1 mg/kg) injection. Saline or CNO was administrated 10 min prior to the entry to the chamber. The side on which saline or CNO was administered was counterbalanced. After three conditioning days, mice were given free access to the entire apparatus and their position was tracked (post-test). The difference of time spent in the CNO zone, the number of entries into the CNO zone and the velocity between the post- and pre- test.

For pup retrieval test (*Figure 7—figure supplement 2K–O*), a home cage of the female subject was used. Female subject was i.p. injected with saline as control or CNO in their home cage. 30 min after administration, three 7-d-old pups were placed in the corners of the home cage. Animal behavior was recorded from the dorsal view for 10 min. Parenting behavior was annotated from the recorded video using Behavioral Observation Research Interactive Software (BORIS) in a frame-by-frame fashion. Pup retrievals, nest building, pup grooming and pup sniffing were annotated based on characteristic postures. Briefly, pup retrieval was defined as a female subject carrying pup. Nest building was defined by female subject digging their bedding or carrying nesting material. Pup grooming was identified by grooming pups without carrying. Female to pup sniffing was identified by female subject nose approaching pup. For neural activation experiments using hM3Dq (*Figure 7—figure supplement 2K-O*), 0.1 mL of 1 mg/kg CNO dissolved in saline or 0.1 mL of saline was I.P. injected into the animal 30 min prior to the pup introduction.

### *In vivo* calcium imaging experiments and analysis

Miniature microscope nVoke 2.0 or nVue (Inscopix) was used to obtain GCaMP6s signal (20 Hz). The microscope was connected above the behavior apparatus using a commutator (Inscopix) to support the weight of the microscope. On the experiment day, the subject was habituated to the microscope for 10 mins. After the habituation, the LED power and gain was adjusted to 0.2–0.5 and 6.0–8.0, respectively depending on the intensity of GCaMP6s expression. Data acquisition started after the adjustment of imaging parameters. The animal was kept habituated to the microscope for additional >5 minto obtain background signal, and then the intruder male animal was introduced to the subject's home cage. Calcium activity as well as animal behavior was recorded until 2–10 min after male ejaculation. Imaging data were spatially down-sampled by factor of 4 and temporally down-sampled

to 10 Hz using the Inscopix Data Processing software (1.8.0). The data was further filtered by a spatial band-pass filter and motion corrected using the same software. Masks were manually drawn to extract data from single cell. The data frame containing GCaMP6s signal intensity traces were exported from the Inscopix Data Processing software and further processed in Python. Signal intensity array was z-scored using the background signal. For peri event activity analysis in *Figures 4 and 5*, z-scored signal was collected from –5 to +15 s around the behavior onset. Peri event activity from multiple behavior events were concatenated and averaged. Finally, the mean value of pre-behavior background (−5–0 s from onset) was subtracted. The magnitude of z-scored signal was quantified by averaging 0 to +5 s from onset. Cells that had larger than 2 d z-scored signal were classified as 'responding' cells. For persistent activity analysis in *Figure 6*, z-scored signal was collected from –5 to +120 s around the onset of male ejaculation. The mean value of pre-behavior background (−5–0 s from onset) was subtracted. The decay length was quantified by calculating the time the z-scored signal was smaller than 0. The magnitude of z-scored signal was quantified by averaging 0 to +5 s from onset. Spectral clustering was conducted using scikit-learn package and previously used custom script (*Namboodiri et al., 2019*). Briefly, first dimensionality reduction using principal component analysis was conducted for z-scored signal from –5 to +120 s around the onset of male ejaculation. Next, n=4 principal components were used to project the data. This was further used as the input into the clustering algorithm while systematically testing nearest neighbor and number of cluster variables. The best variable that resulted in maximizing the silhouette score was determined (number of clusters = 5, number of nearest neighbors = 55, average silhouette score = 0.405). All clusters contained cells from multiple animal subjects, indicating the variability of the dataset can be explained by heterogenous cell population, rather than subject variability.

## TRAP experiments

TRAP method was conducted as previously described with modifications (*DeNardo et al., 2019*). TRAP2::Ai14 or TRAP2 mice were OVX during isoflurane anesthesia and AAV injected if necessary. After 2–3 wk from OVX surgery, animals were prepared for the activity labeling assay. Female subjects were subcutaneously (s.c.) injected with 0.1 mL and 0.05 mL Estradiol (E8515, Sigma-Aldrich, 0.1 mg/ml) dissolved in sesame oil (Sigma-Aldrich) at 24 and 48 hr before assay. On the day for the assay, female subjects were isolated in their home cage and introduced with a male mating partner to either experience appetitive behaviors, consummatory behaviors, or male ejaculation. For the appetitive group, the male partner was removed after 10–20 min. To ensure that the female subject did not experience any consummatory behavior from the male, the male partner was removed whenever an attempt to mount was observed. For the consummatory group, the male partner was removed after the first male mounting and intromission was observed. For the post-ejaculatory group, the male partner was removed just after male ejaculation. Immediately after the removal of the male animal, the female subject was I.P. injected with 0.125 mL of 50 mg/kg of 4-Hydroxytamoxifen (Sigma-Aldrich, cat#H6278) dissolved in sesame oil. After the injection, the female subject was isolated in their home cage and kept in the behavior room for overnight to avoid background stimulation. On the following day, the mice returned to the regular housing.

## HCR experiments and analysis

Hybridization chain reaction (HCR) was performed based on the protocol provided by Molecular Instruments with some modifications (*Figure 3*, *Figure 7—figure supplement 1D-G*). For *Fos* activity analysis (*Figure 3*), the female subject was first singly housed in the home cage for 1 hr. The male partner was introduced into the home cage and animal behavior was recorded. For the appetitive group, the male partner was removed after 10–20 min until they first attempt mounting to the female subject. For the completion group, the male partner was removed just after male ejaculation. Twenty-minutes after the removal of the male partner, the female subject was immediately sacrificed to collect brain tissue. The brain tissue was then snap-frozen using dry ice. Every fifth 20 micron coronal sections containing the MPOA (three sections from Bregma + 0.10 mm to –0.10 mm) were collected on a slide glass and fixed with 4% PFA in PBS for 30 min and rinsed with PBS, followed by dehydration with a series of 50%, 70%, 100%, 100% of ethanol. After final dehydration, the slides were dried. The tissue was treated with Protease4 (ACDBiosystem) for 5–7 min. The slides were then rinsed with PBS and were applied with hybridization buffer (Molecular Instruments) for 10 min at 37°

C for pre-hybridization. RNA probes for *Fos, Slc32a1* (Vgat), *Slc17a6* (Vglut2), *Calcr, Esr1, Gal, Nts, Prlr* (1 mM, Molecular Instruments) were diluted in a ratio of 1:100–1:250 into the hybridization buffer. After pre-hybridization, hybridization buffer (70 µl) containing the probes was applied to each slide on which the parafilm cover was placed. After 12h–18h of incubation at 37° C in the moisture chamber, the sections were washed, with a series of probe wash buffer and SSC-T buffer (5 x SSC, 0.1% Tween 20) mixture. After the final wash, the sections were incubated in amplification buffer (Molecular Instruments) for 30 min at room temperature for pre-amplification. Amplification hairpin probes (3 mM, Molecular Instruments) were diluted in a ratio of 1:50 into the hybridization and then snap cooled for 1.5 min in 95° C. After pre-amplification, amplification buffer containing hairpin probes (70 µl) was applied to each slide on which the parafilm cover was placed. After 12 hr–18 hr of incubation at room temperature in the moisture chamber, the sections were washed in 5x SSC-T buffer for 30 min 2x. The sections were quenched for autofluorescence using Vector TrueView reagent (Vector, cat# SP-8400). After washed in 2x SSC, the sections were stained with DAPI in PBS (5 µg/ml, Thermo Fisher Scientific, cat#D1306) for 8 min. The slide was then mounted using Vibrance Antifade Mounting Medium (VECTASHIELD, cat#H-1800). Images were obtained using a conventional microscope (Zeiss, Apotome Imager.M2) with a 20 x objective. After image acquisition, the coverslip was removed and washed in 2 x SSC for 10 min, then incubated with DNase1 (0.25 U/µl, Roche, cat#4716728001) for 1.5 hr at room temperature. The section was then washed in 2 x SSC for 5 min x6. The sections then proceeded to the pre-hybridization step for the next round of HCR. A total of three rounds of HCR were conducted. Images from multiple rounds were aligned using HiPlex image registration software (ACDBiosystem) or manually aligned using ImageJ package TrackEM2 (*Cardona et al., 2012*).

For anatomical characterization for TRAP2 animals used in *Figure 7—figure supplement 1D–G*, subjects were perfused with PBS and 4% PFA-PBS, then brain was harvested and post-fixed with PFA overnight. Every fourth 40 µm coronal sections containing the MPOA were collected on a slide glass. The sections went through one round of HCR and were stained for Vgat and Vglut2 following the procedures described above.

The images with each probe signal were aligned to the DAPI channel to identify MPOA subregions and adjacent regions. The image was processed in HALO v3.2 software (Indica Labs) to segment each cell based on the DAPI signal and to quantify RNA transcript signal in each cell. Parameters for detecting puncta and intensity of each gene were manually adjusted for each tissue section of each animal by a volunteer blind to the experimental condition. The cutoff value to determine a gene 'expressing cell' was defined as the mean fluorescent intensity of 3–5 transcripts depending on the background fluorescent signal. For supervised clustering of the MPN cell gene expression matrix (*Figure 3K–M*) was conducted using Seurat3.1.1 in R (*Stuart et al., 2019*). Cells that had no mRNA signal were excluded from the dataset. This expression matrix contained single cells detected across all 10 animals and the spot counts for every gene; we also included signal intensity measurements as metadata. Once loaded, the HCR digital expression matrix was scaled but not normalized. During scaling, we regressed out gene intensity measurements on the basis that signal intensity may indicate staining quality. Next, cells were clustered using the same pipeline. The cells were UMAP plotted using nearest_neighbors = 100, minimum_distance = 0.05 (*Figure 3K*). The cluster ID for each cell was exported and used for statistical analysis using custom Python scripts.

## Immunohistochemistry experiments

To evaluate the accuracy and efficiency of TRAP labeling, we utilized subjects used in pharmacogenetic experiments (*Figure 7*, *Figure 7—figure supplement 1*) and labeled neurons that respond to male ejaculation using immunohistochemistry against immediate early gene c-Fos. The TRAP2 female subjects were labeled with eYFP or hM3Dq-mCherry in the appetitive or male ejaculation responding neurons in the MPOA (appetitive-hM3Dq, consummatory-hM3Dq, post-ejaculatory-hM3Dq or post-ejaculatory-eYFP group). Ninety-minutes before perfusion, the female subjects experienced male ejaculation during a home cage mating assay (described in 'Behavior experiments and analysis.') to activate the male ejaculation responding neurons in the MPOA. The subjects were then perfused with PBS and 4% PFA-PBS, and the brain was harvested and post-fixed with PFA overnight. Every fourth 40 µm coronal sections containing the MPOA were collected and washed three times with PBS-T (1 x PBS, 0.3% Triton-X, Sigma-Aldrich, cat# 11332481001) for 10 min x3, and treated with 5% normal donkey serum (NDS, Sigma-Aldrich, cat# 566460) in PBST for 1 hr at room temperature

for blocking. Sections were then incubated with rabbit anti-cFos antibody (Abcam, cat#190289-Rb, 1:500), diluted into 5% BSA in PBST for overnight at 4° C. Antibody signals were detected by goat anti-rabbit Alexa 488 or Alexa 594 (Invitrogen, cat#A21206 or cat#A21209, 1:500) for 2 hr at room temperature. Sections were washed once with PBST for 10 min, treated with PBS containing DAPI for 20 min, rinsed with PBS, and mounted with cover glass using Vibrance Antifade Mounting Medium (VECTASHIELD, cat#H-1800).

Sections were imaged by FV-3000 confocal microscope (Olympus) with a 20 x objective. Images were processed in cellpose to segment DAPI-positive cells in the MPOA and their subregions (*Stringer et al., 2021*). DAPI positive cell masks were used to quantify intensity of c-Fos protein or YFP/mCherry signal. The intensity value was used to determine the negative and positive cell.

## Brain-wide analysis of activity-dependent labeling experiments

TRAP2::Ai14 females were used to label neurons responding to appetitive behavior or male ejaculation. Two-weeks after activity labeling (described in 'TRAP experiments.'), the animals were perfused with PBS and 4% PFA-PBS, the brain was harvested and post-fixed with PFA overnight. After post-fixation, the brains were processed for tissue clearing. For tissue clearing, we utilized stabilization under harsh conditions via intramolecular epoxide linkages to prevent degradation method (SHIELD) (*Park et al., 2019*). The procedures followed the SHIELD tissue clearing protocol (Full Active Pipeline Protocol v5.05, LifeCanvas). Briefly, the brain was incubated in SHIELD OFF solution [2.5 mL DI water, 2.5 mL SHIELD-Buffer Solution, 5 mL SHIELD-Epoxy Solution] at 4° C for 4 d. Then transferred to SHIELD ON buffer and incubated at 37° C for 24 hr. Following, SHIELD ON buffer incubation, the brain was pre-incubated in Delipidization buffer for overnight at room temperature. The tissue was then placed into SmartClear II Pro (LifeCanvas) to conduct active tissue clearing for 24 hr. To match the refraction index, the tissue was incubated in 50% of EASYINDEX solution (RI = 1.52, LifeCanvas) at 37° C shaking for 24 hr and then 100% of EASYINDEX solution at 37° C shaking for 24 hr. The tissue was then embedded in EASYINDEX with 2% agar and mounted onto SmartSPIM light sheet microscope (LifeCanvas). The tissue was mounted on the ventral side facing toward the objective, imaged from the horizontal view using a 3.6 x objective with 4 µm zstep. 594 nm channel was collected as the signal channel, 488 or 647 nm channel was collected as the background channel. Background channel was used for the alignment to the brain atlas. The images were first destriped and then stitched using a custom written script (LifeCanvas). The images were pre-processed and down sampled by two times (final resolution (x,y,z) = (3.6 µm, 3.6 µm, 4 µm)). Images were registered to a standard brain atlas Unified Anatomical Atlas (Yongsoo Kim Lab) (resolution (x, y, z) = (20 µm, 20 µm, 50 µm)) using ClearMap pipeline on a cloud-based cluster computer (*Chon et al., 2019*; *Madangopal et al., 2022*; *Renier et al., 2016*). tdTomato expressing cells were segmented through the same pipeline using a classifier created using Ilastik (*Berg et al., 2019*). To train the classifier, 10 chunks of images from the four of the subjects were randomly sampled and used as the training dataset (Image size, (x, y, z)=1.8 mm × 1.8 mm 100 µm). One chunk image of the same size from each subject was used as the test dataset. The quality of the classifier was visually examined. The total number of tdTomato positive cells per brain region was quantified and normalized by the size of each brain region (density, mm³). In the standard brain atlas, brain regions were hierarchically sorted from broader regions to finer regions (level 1 to level 8). Level 4 was selected for broader analysis of brain regions (*Figure 2C*). Level 6 was selected for finer analysis of brain regions (*Figure 2H*, *Figure 2—figure supplement 2*). To compare the means of density of tdTomato cell in each brain area between the appetitive, consummatory and post-ejaculatory group, we conducted an ANOVA test and further corrected for multiple comparisons with Benjamini/Hochberg correction. False discovery rate (FDR) = 0.05 was used for the broader analysis while a loose discovery rate (FDR = 0.10) was used for finer analysis due to the large number of comparisons. A post-hoc Tukey's HSD post-hoc test (FWER = 0.05) was conducted for regions that tested significant in the ANOVA test.

## Quantification, data processing, and statistical analysis

Data were presented as mean with error bar of 95% confidence interval unless otherwise mentioned. Data processing and plot generation was done in Python and R. Plots were further processed in Adobe

Illustrator to create Figures. The sample sizes were chosen based on previous studies using similar behavioral experiments. Codes used for analysis and statistics are provided on GitHub (https://github.com/stuberlab/Ishii-MPOA-Ejaculation-2023). The statistical details of each experiment including the statistical tests used, exact value of n and what n represents, are shown in each figure legend. Each statistical test result is shown in *Supplementary file 1*. Significance was shown as *p<0.05, **p<0.01, ***p<0.001 and not significant values were not noted or shown as n.s.

The sample size for behavior experiments in this study were n=5–9. This was predetermined based on previous studies examining female sexual behavior (*Ishii et al., 2017*; *Liu et al., 2022*; *Yin et al., 2022*). To further examine the number of animals required for our behavioral experiments, we pooled data used in this study and conducted a power analysis (n=111 pooled data, control n=94, stim n=17). We conducted a power analysis using the variance calculated from pooled average time in isolation zone. These data were pooled from control animals in each experiment (eg. animals with GFP control virus injected, saline injected, etc.). The average time in isolation zone in the after ejaculation or after reactivating the completion cells was 420±210 s, and 49±91 s in the control group (mean ± s.d.). Within this population, we found that five animals were sufficient to detect the difference (p<0.05, power = 0.8) in Students t-test.

## Acknowledgements

This work was supported by a JSPS oversea fellowship (KKI), Uehara postdoctoral fellowship (KKI), NIH grants R37 DA032750, R01 DA038168, R01 DA054317, P30 DA048736, and Washington Research Foundation Postdoctoral Fellowship (ERS).

## Additional information

### Funding

| Funder | Grant reference number | Author |
| --- | --- | --- |
| National Institute on Drug Abuse | DA032750 | Garret D Stuber |
| National Institute on Drug Abuse | DA038168 | Garret D Stuber |
| National Institute on Drug Abuse | DA054317 | Garret D Stuber |
| National Institute on Drug Abuse | DA048736 | Garret D Stuber |

The funders had no role in study design, data collection and interpretation, or the decision to submit the work for publication.

### Author contributions

Kentaro K Ishii, Conceptualization, Data curation, Formal analysis, Investigation, Visualization, Writing - original draft; Koichi Hashikawa, Investigation, Writing – review and editing; Jane Chea, Rebecca Erin Fox, Data curation, Investigation, Writing – review and editing; Shihan Yin, Suyang Kan, Meha Shah, Data curation, Writing – review and editing; Zhe Charles Zhou, Jovana Navarrete, Alexandria D Murry, Eric R Szelenyi, Methodology, Writing – review and editing; Sam A Golden, Resources, Methodology, Writing – review and editing; Garret D Stuber, Conceptualization, Resources, Supervision, Funding acquisition, Writing - original draft, Writing – review and editing

### Author ORCIDs

Kentaro K Ishii  http://orcid.org/0000-0002-7956-0806
Garret D Stuber  https://orcid.org/0000-0003-1730-4855

### Ethics

All experiments were approved by the Institutional Animals Care and Use Committee at the University of Washington (Protocol #4450-01).

Reviewer #1 (Public review): https://doi.org/10.7554/eLife.91765.3.sa1
Reviewer #2 (Public review): https://doi.org/10.7554/eLife.91765.3.sa2
Reviewer #3 (Public review): https://doi.org/10.7554/eLife.91765.3.sa3
Author response https://doi.org/10.7554/eLife.91765.3.sa4

## Additional files

### Supplementary files

Supplementary file 1. All results from statistical analysis, except for ones related to *Figure 2*, are included in Supplementary file 1.

Supplementary file 2. Related to *Figure 2C*. One-way ANOVA test with Benjamini/Hochberg correction (FDR = 0.05).

Supplementary file 3. Related to *Figure 2I*, *Figure 2—figure supplement 2*. One-way ANOVA test with Benjamini/Hochberg correction (FDR = 0.10) followed by post-hoc Tukey's HSD test (FWER = 0.05).

MDAR checklist

### Data availability

Whole brain FosTRAP activity mapping data is available on DANDI (https://doi.org/10.48324/dandi. 001249/0.241109.2357). All data used in this project is shared on Figshare (https://doi.org/10.6084/ m9.figshare.27642768.v1). The code for analysis and data generation is shared on GitHub (https:// github.com/stuberlab/Ishii-MPOA-Ejaculation-2023 copy archived at *Stuber Lab, 2025*).

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
