## [Editor Report · eLife Assessment]

This **important** work combines molecular genetics and behavioral analyses to identify inhibitory neurons in the female medial preoptic area as a neural locus that is activated following male ejaculation and whose prolonged activity plays a key role in the regulation of female sexual motivation. These experiments are rigorous and well-performed. The data are **compelling** and demonstrate that a subpopulation of neurons in the medial preoptic area are selectively activated following the completion of mating in females. The medial preoptic area has long been implicated as critical to sexual behavior in both sexes; however the use of a self-paced mating assay for females provides fine control over manipulating and monitoring cellular activity in this region during more naturalistic behavior. In addition, this study may act to inspire others to further explore the additional brain regions found to show upregulation of neural activity (Fos) during mating completion in females using the datasets generated here.

---

## [Referee Report · Reviewer #1 (Public review)]

Summary:

This manuscript by Ishii et al utilize a classical, but extremely understudied, female self-paced assay to directly address aspects of female sxual motivation independent from the male's behavior. This allowed for clear separation of appetitive and consummatory events, of which whole brain unbiased activity was mapped. Mating completion in females was then focused to the medial preoptic nucleus where the authors performed a rigorous set of single-cell GCaMP recordings in populations marked by Vglut2 and Vgat, finding the latter display stronger and prolonged activity after the onset of mating completion. Finally, they demonstrate function to these Fos-TRAPPED completion cells demonstrating their capacity to suppress female sexual behavior.

Strengths:

This manuscript sought to explicitly explore female mating drive as dictated by the female, a very rare angle for those studying mating behavior which almost always is controlled by the male's behavior. To achieve this, the authors went back to old literature and modified a classical paradigm in which measurable approach and avoidance of male conspecifics can be measured in female mice using a self-paced mating assay. Strengths include a detailed quantification of female behaviors demonstrating a robust attenuated sexual motivation in females after mating completion. To determine the neural basis behind this, a brain wide analysis of cells responding to mating completion in the female brain was conducted which revealed numerous anatomical regions displaying increased Fos activity, including the MPOA, of which the authors concentrated the remaining of their study. Employing microendoscopic imaging, the authors discovered that this mating completion signal was strongly represented in the MPOA. The single cell data analyses are of very high quality as is the number of individual cells resolved. While they identified both excitatory and inhibitory cell types that were activated by mating completion, they found the latter exhibited stronger and more persistent activity. Segmentation into individual mating behaviors reinforced the importance of GABAergic completion cells, which display prolonged activity late after the onset of mating completion. This information provides a potential mechanism for how female mice suppress further mating activity following completion. The authors then definitively demonstrate this function by TRAP'ping completion cells with chemogenetic actuators and show that CNO-induced activation of these cells specifically and strongly suppresses female sexul behavior. All experiments were extremely well-designed and performed carefully and expertly with the necessary controls solidifying the conclusions.

Weaknesses:

While there are no glaring weaknesses in this study, it should be noted that a great deal of literature has pinpointed the MPOA (and specifically inhibitory cells in this area) as being critical to sexual behavior, including female mating. However, no study to my knowledge has explored self-paced female mating with such fine control over manipulating and monitoring cellular activity in this region. In addition, this study may act to inspire others to further explore the additional brain regions found to show upregulation of neural activity (Fos) during mating completion in the female using the data sets generated here.

Comments on revisions: The data has been provided in a public database.

---

## [Referee Report · Reviewer #2 (Public review)]

Summary:

In this set of studies, authors identify cFos activation in neurons in female mice that mated with males, and after experiencing male sexual behavior that is either restricted to appetitive behavior or including ejaculation. The medial preoptic nucleus was identified as an area with high cFos induction following ejaculation. Characterization of neurochemical phenotypes of cfos-expressing neurons showed a heterogenous distribution of activated neurons in the MPOA, including both inhibitory and excitatory cell types. Next, *in vivo* calcium imaging was used to show activation of Vgat and Vglut neurons in female mice MPOA after displaying sniffing of the male, experiencing male appetitive, or male consummatory sexual behavior, demonstrating significantly higher activation and of a greater subpopulation of Vgat neurons than Vglut neurons. Moreover, greatest activation of Vgat neurons was detected following experiencing ejaculation, and ejaculation activated different subpopulations of MPOA cells than consummatory or appetitive sexual behaviors experienced by the female. Finally, pharmaco-genetic activation of the subpopulation of MPOA neurons that were previously activated following ejaculation resulted in a significant reduction of approach behavior by the female mice towards the male, interpreted as suppression of female sexual motivation. In conclusion, a subpopulation of inhibitory cells in the MPOA is activated in female mice after experiencing ejaculation, in turn contributing to suppression of sexual approach behavior.

Strengths:

The current set of studies replicates previous findings that ejaculation causes longer latencies to initiate interactions with a male after receiving an ejaculation in a paced mating paradigm, which is widely validated and extensively used to investigate sexual behavior in female rodents. Studies also confirm that ejaculation increases cFos expression in the MPOA, while extending prior findings with a careful analysis of the neurochemical phenotype of activated neurons. A major strength of the studies is the use of cell-specific *in vivo* imaging and pharmaco-genetic activation to reveal a functional role of specific neuronal ensemble within the MPOA for post ejaculatory female sexual behavior.

Weaknesses:

The authors include an elegant manipulation of ejaculation-activated neurons in the MPOA using DREADD. However, this study was limited to show that activation of previously activated cells was sufficient to reduce approach behavior in a paced mating paradigm and receiving intromissions in a home cage mating paradigm. An inhibition approach using DREADD would have been a great complement to this study as it would have examined if activation of the cells was required. Moreover, additional tests for sexual motivation would have greatly strengthened the overall conclusions.

---

## [Referee Report · Reviewer #3 (Public review)]

Summary:

Ishii et al used molecular genetics, behavioral analyses, *in vivo* neural activity imaging, and neural activity manipulations in mice to study the functional role of a subset of medial preoptic area (MPOA) neurons in the regulation of female sexual drive. They first employed a self-paced mating assay during which a female could control the amount of interaction time with a male to assess female sexual drive after completion of mating. The authors observed that after mating completion (i.e., male ejaculation) females spend significantly less time interacting with males, indicating that their sexual drive is reduced. Next, the authors performed a brain-wide analysis of neurons activated following male ejaculation and identified the MPOA as a strong candidate region. One caveat is that the activity labeling was not exclusive to neurons activated following male ejaculation but included all neurons activated before, during, and after the mating encounter. However, in this revised version of the manuscript, the authors have included a key control group that labels all neurons activated up to but not including male ejaculation. Comparison of the number of activated neurons in these two groups revealed a significant additional set of neurons in the female MPOA following ejaculation. Importantly, the authors also provided *in vivo* calcium imaging data showing that a subset of MPOA neurons responds significantly and specifically to male ejaculation and not other behaviors during the social encounter. The authors performed these studies in both excitatory and inhibitory populations of the MPOA. Their analysis identified a subpopulation of inhibitory neurons that exhibit sustained increased activity for 90 sec following male ejaculation. Finally, the authors used chemogenetics to activate MPOA neurons during home cage mating, condition place preference, pup retrieval, and the self-paced mating assay. They found that activation of female MPOA neurons that were previously activated following male ejaculation significantly reduces mating behaviors and time spent interacting with a male during the self-paced mating assay. Whereas, activation of female MPOA neurons that were previously activated during consummatory behaviors but not male ejaculation does not alter mating behaviors and time spent interacting with a male. Therefore, MPOA neurons activated following ejaculation are sufficient to suppress female sexual motivation.

The authors' experimental execution is rigorous and well performed. Their data identify inhibitory neurons in the female MPOA as a neural locus that is activated following male ejaculation and whose prolonged activity plays a key role in the regulation of female sexual motivation. The addition of some key control groups to this revised version of the manuscript greatly strengthens the interpretation of the authors' findings.

Strengths:

(1) The use of the self-paced mating assay in combination with neural imaging and manipulation to assess female sexual drive is innovative. The authors correctly assert that relatively little is known about how male ejaculation affects sexual motivation in females as compared to males. Therefore, the data collected from these studies is important and valuable.

(2) The authors provide convincing histological data and analyses to verify and validate their brain-wide activity labeling, neural imaging, and chemogenetic studies.

(3) The single cell *in vivo* calcium imaging data are well performed and analyzed. They provide key insights into the activity profiles of both excitatory and inhibitory neurons in the female MPOA during mating encounters. The authors identification of an inhibitory subpopulation of female MPOA neurons that is selectively activated following completion of mating is fundamental for future experiments which could potentially find a molecular marker for this population and specifically manipulate these neurons to understand their role in female sexual motivation in greater detail.

(4) The authors provide convincing evidence that activation of female MPOA neurons activated following male ejaculation is sufficient to suppress female sexual motivation. Importantly, the authors addition of the consummatory-hM3Dq group demonstrates that activation of female MPOA neurons activated during mating behaviors prior to male ejaculation is not sufficient to suppress female sexual motivation.

Weaknesses:

In this revised version of the manuscript, the authors have added important controls as well as additional clarifying text that adequately address the weaknesses that were present in the original version of the manuscript.

---

## [Author Response]

The following is the authors’ response to the original reviews.

**Public Reviews:**

**Reviewer #1 (Public Review):**
[…] Weaknesses:While there are no glaring weaknesses in this study, it should be noted that a great deal of literature has pinpointed the MPOA (and specifically inhibitory cells in this area) as being critical to sexual behavior, including female mating. However, no study to my knowledge has explored self-paced female mating with such fine control over manipulating and monitoring cellular activity in this region. In addition, this study may act to inspire others to further explore the additional brain regions found to show upregulation of neural activity (Fos) during mating completion in the female using the data sets generated here.
**Reviewer #2 (Public Review):**
[…] Weaknesses:The authors include an elegant manipulation of ejaculation-activated neurons in the MPOA using DREADD. However, this study was limited to show that activation of previously activated cells was sufficient to reduce approach behavior in a paced mating paradigm and receiving intromissions in a home cage mating paradigm. An inhibition approach using DREADD would have been a great complement to this study as it would have examined if activation of the cells was required. Moreover, additional tests for sexual motivation would have greatly strengthened the overall conclusions.
**Reviewer #3 (Public Review):**
[…] Weaknesses:(1) Their activity-dependent labeling strategy is not exclusive to mating completion but instead includes all neurons active before, during, and after the social encounter. In the manuscript, the authors did not discuss the time course of Fos activation or the timeframe of the FosTRAP labeling strategy. Fos continues to be expressed and is detectable for hours following neural activation. Therefore, the FosTRAP strategy also labels neurons that were activated 3 hours before the injection of 4-OHT. The original FosTRAP2 paper which is cited in this manuscript (DeNardo et al, 2019) performed a detailed analysis of the labeling window in Supplementary Figure 2 of that paper. Here is quoted text from that paper: "Resultant patterns of tdTomato expression revealed that the majority of TRAPing occurred within a 6-hour window centered around the 4-OHT injection." Thus, the FosTRAP "mating completion" groups throughout this manuscript also include neurons activated 3 hours before mating completion, which includes neurons activated during appetitive and consummatory mating behaviors.This makes all of the FosTRAP data very difficult to interpret. Compounding this is the issue that the two groups the authors compare in their experiments are females administered 4-OHT following appetitive investigation behaviors (with the male removed before mating behaviors occurred) and females administered 4-OHT following mating completion. The "appetitive" group labeled neurons activated only during appetitive investigation, but the "completion" group labeled neurons activated during appetitive investigations, consummatory mating bouts, and mating completion. Therefore, in the brain-wide analysis of Figure 2, it is impossible to identify brain regions that were activated exclusively by mating completion and not by consummatory mating behaviors. This could have been achieved if the "completion" group was compared to a group of females that had commenced consummatory mating behaviors but were separated from the male before mating was completed. Then, any neurons labeled by the "completion" FosTRAP but not the "consummatory" FosTRAP would be neurons specifically activated by mating completion. In the current brain-wide analysis experiments, neurons activated by consummatory behaviors and mating completion can not be disassociated.This same issue is present in the interpretation of the chemogenetic activation data in Figure 6. In the experiments of Figure 6, the authors are activating neurons naturally activated during consummatory mating behaviors as well as those activated during mating completion.

We appreciate the reviewers comments and concerns about the TRAP method.

First, we agree that the FosTRAP method does not have the sensitivity to separate ensembles that happen within a short time window. From our preliminary results, we have observed that the cells that inject 4-OHT after mating completion induce more tdTomato cells in the MPN than injection after appetitive behavior or consummatory behavior (Author response image 1).

To further compare the difference between the “consummatory” and “completion” ensemble, we included an additional cohort where we TRAP cells responding to consummatory behavior. This cohort is added to Figure 2, 6, S3, S4, S9, S10 and S11. From the whole brain mapping of TRAP cells, we found that many hypothalamic and extended amygdala areas including the medial preoptic area, and the bed nucleus of stria terminalis were shown to have significantly larger tdTomato+ cell density in the completion group than in the appetitive group while there was a tendency that the consummatory group also had larger cell density than the appetitive group. In the Gq-DREADD experiment, we found that the Completion-hM3Dq group but not the Consummatory-hM3Dq group showed the reduction of sexual motivation of the female mouse in the self-paced mating assay (Figure 6). The Completion-hM3Dq group but not the Consummatory-hM3Dq group also showed significantly low intromission events and tended to show lower receptivity in the home cage mating assay (Figure S10). Furthermore, post-hoc histological analysis also showed that the c-Fos+ and TRAP labeled cells in the MPN tended to be the larger in the Completion-hM3Dq group than in the Consummatory-hM3Dq group (Figure S9). These results, together with the *in vivo* Calcium imaging experiments in Figure 3, 4 and 5, suggests that the MPN contains male-ejaculation responsive cells that are distinct with the male-mounting responsive cells and that they are sufficient to suppress female sexual motivation.

However, it is true that with the current state of mouse genetic tools, we do not have any methods with higher time accuracy. We have discussed the limitations of FosTRAP method regarding its low time sensitivity in the Discussion section.

**Author response image 1. sa4fig1:** Representative image showing TRAP labeling in the MPN after mating completion and intromission.

(2) This study does not definitively show that the female mice used in this study display decreased sexual motivation after the completion of mating. The females exhibit reduced interaction with males that had also just completed mating, but it is unclear if the females would continue to show reduced interaction time if given the choice to interact with a male that was not in the post-ejaculatory refractory period. Perhaps, these females have a natural preference to interact more with sexually motivated males compared to recently mated (not sexually motivated) males. To definitively show that these females exhibit decreased sexual motivation the authors should perform two control experiments: (1) provide the females with access to a fully sexually motivated male after the females have completed mating with a different male to see if interaction time changes, and (2) compare interaction time toward mated and non-mated males using the self-paced mating assay. These controls would show that the reduction in the interaction time is because the females have reduced sexual motivation and not because these females just naturally interact with sexually motivated males more than males in the post-ejaculatory refractory period.

We highly appreciate the reviewers comments regarding the interpretation of the self-paced mating assay. To address the concerns, we added an experiment where the female subjects were introduced to a novel sexually motivated male mice in the self-paced mating assay immediately after receiving ejaculation (Figure S2). As result, we found that similar to the self-paced mating assay using the same male animal, the female subject spends significantly more time in the isolation zone on the post-ejaculation day when compared to the pre-ejaculation day.

(3) It is unclear how the transient 90-second response of these MPOA neurons following the completion of mating causes the prolonged reduction in female sexual motivation that is at the minutes to hours timeframe. No molecular or cellular mechanism is discussed.(4) The authors discuss potential cell types and neural population markers within the MPOA and go into some detail in Figure S3. However, their experiments are performed with only the larger excitatory and inhibitory MPOA neural populations.

While the molecular or cellular mechanism of prolonged activity of MPOA neurons is critical to understand the neural mechanism of how sustained neural activity in the MPOA suppress female sexual motivation, it is out of the reach of the current manuscript and a subject of future studies. We have added a section in the discussion part to further discuss the potential molecular mechanisms.

**Recommendations for the authors:**

**Reviewer #1 (Recommendations For The Authors):**
If the authors haven't already, it would be useful if the authors could make the brain-wide analysis of Fos activity publicly available.

We have distributed the data to https://dandiarchive.org/

I would also make sure the n's are included in each Figure Legend for each panel (some are missing in the Supplementals).

We appreciate the comment, we have added the number of subjects to Figure 3, 4, 5.

It would also be best to provide clearer labels to some of the Figures, for example, Figure 5D, the Types should also be labeled with what behaviors they correspond to.

We appreciate the comment. Figure 5 is focused on post-ejaculation neural activity. The cell types are categorized based neural activity after experiencing male ejaculation, it does not correspond to any behaviors.

**Reviewer #2 (Recommendations For The Authors):**
(1) A first recommendation is to replace the use of the term "mating completion" with "ejaculation". Male and female rodents display a period of reduced approach behavior following display or experiencing ejaculation, which is referred to as the post-ejaculatory interval. The current studies investigate the neural ensemble that contributes to this post-ejaculatory interval in female mice. In addition, male and female rodents will display a prolonged period of sexual inactivity referred to as satiety, which is typically observed after repeated display or experience of ejaculations. The current studies do not investigate satiety. Moreover, in the current studies, female mice appeared to display approach behavior (time in the interaction zone) even within the 10 minutes following experiencing ejaculation (Fig 1F). Hence, the term "completion" is not accurate and should be replaced by "ejaculation" in all figures and throughout the manuscript. Replacing completion with ejaculation will also clarify what defines "onset of completion", which this reviewer assumes refers to the onset of ejaculatory behavior observed in the male.

Thank you for the comment. We agree that the mating completion was inappropriate. We have changed the wording to ejaculation or post-ejaculatory period.

(2) Likewise, a variety of other terms and descriptions need to be adjusted for consistency and accuracy. For example, "room" when referring to the interaction or isolation zones; "onset of mating completion" when referring to ejaculation; "male intruder" to refer to the introduction of the male mating partner, but using a term typically used for an intruder-resident aggression test. Replacing these terms will aid in reducing confusion for the reader and more accurately describe the behavioral parameters.

We appreciate the comment. We have updated the terms “male intruder” to “partner”, “room” to “area” or “zone”.

(3) The use of the paced mating paradigm is a strength of these studies. This paradigm has been widely used and validated to study female sexual behavior in rodents. Please refer to recent reviews and landmark papers using this paradigm in addition to the current cited papers to better reflect the vast wealth of studies that previously reported the behavioral data that were replicated in this study.

We have added a section discussing the self-paced mating assay, its merits and caveats P8.

(4) In the paced mating test, females can pace the receipt of sexual stimulation, and latencies to withdraw and return to the male-containing chamber are considered indicators of sexual motivation. Female withdrawal will increase with the intensity of the sexual stimulation and latency to return is longer following ejaculation. Paced mating is thus a balance of approach and withdrawal behaviors that increases reward and likelihood of pregnancy for females. Moreover, ejaculation-induced withdrawal and longer latencies to return and approach are altered by hormonal status and by the introduction of a novel male partner. Thus, female sexual behavior is complex and withdrawal behavior (in this paper measured as time spent in an isolation zone) needs to be interpreted with caution and not simply referred to as sexual motivation. I recommend expanding the description of the paradigm to highlight the strengths and limitations of this paradigm and use caution to interpret time spent in the isolation zone as a lack of sexual motivation. I also recommend referring to the period after ejaculation as the post-ejaculatory interval (instead of completion).

Thank you for the comment. We have changed the wording in the manuscript to adjust the way it refers to sexual motivation.

(5) In the current paper, time in the isolation zone and the number of transitions are used as the behavioral measures. Latencies, which are typically included in paced mating studies, were missing from the data. If data are available for latencies to withdraw and return to the interaction zone after mount, intromission, and ejaculation, please add these data. If such data were not collected or are not available, please recognize this caveat.

Thank you for the comment. In figure 1, which all animals did experience male ejaculation, we added latency analysis (Figure 1I and 1P). The result indicates as suggested in the literature, female mice took significantly longer to return the interaction zone after male-ejaculation.

(6) The brain-wide mapping study of cFos expression after ejaculation confirms and extends prior findings, mostly in rats. Please reference prior papers in female rodents showing cFos after ejaculation and discuss how the current data replicate or differ from prior data.

In the manuscript P8 L351, we have referred to Pfaus et al., 1993 to discuss the similarity in the c-Fos expression pattern studied in rats. We have further added descriptions to emphasize the similarity between the two datasets.

(7) A paragraph describing the specific cell types that are activated in the MPOA is an essential part of the study and is described in detail, but only shown in supplementary figures. Given the emphasis on this particular part of the study, a recommendation is to incorporate these data as a regular figure instead of supplementary material.

While we greatly appreciate the comment, we consider that the molecular characterization of MPOA neurons are not the main focus of the paper and decided to keep it in the supplementary figure.

(8) Calcium imaging studies were performed in the home cage for obvious practical reasons. However, in the home cage testing, the females withdraw from the males using a different approach and do not exit an interaction zone through a division. There may also be differences in the male sexual behavior patterns and thus the stimulation that females receive from the male. Yet, it appears that ejaculation induces similar patterns of neural activation in this paradigm. Thus, it is likely that neuron activation is a result of receiving ejaculation, rather than withdraw behavior. Please briefly discuss the comparisons between the cFos and calcium imaging conclusions in these two different paradigms.

We have added a section discussing the self-paced mating assay, its merits and caveats P8. Withdrawal and latency and its interpretation is discussed in this section.

(9) The final study includes the manipulation of ejaculation-activated neurons in the MPOA using DREADD. This study was limited to show that activation of previously activated cells was sufficient to reduce approach behavior in a paced mating paradigm and receiving intromissions in a home cage mating paradigm. An inhibition approach using DREADD would have been a great complement to this study as it would have shown if activation of the cells was required. Moreover, additional tests for sexual motivation, such as partner preference tests would have greatly strengthened the results since a lack of entering an interaction zone can also be explained by impaired sensory processing or locomotor behavior. Finally, CNO also appeared to impact time in the isolation zone for a subset of animals in the ejaculation (completion) control group and the appetitive group. These effects didn't reach statistical significance, but groups also had low sample sizes (n=6-7) and may thus have been underpowered. The recommendation is to include these caveats and shortcomings in the discussion of these results.

We appreciate the comments. We first added an inhibitory approach to show the necessity of MPOA neurons. As result, we found that the inhibition of these neurons did not affect the behavior in the self-paced mating assay but increased the subjects sexual receptivity (Figure S11). For the low sample size, we have added a power analysis in the statistical section.

(10) The studies utilized ovariectomized females with hormone priming. Since sexual receptivity in females is highly dependent on the hormonal milieu, the authors are encouraged to add an explanation of why ovariectomized females were used and if the results may have differed in cycling females.

We appreciate the comments. The female subjects used in the TRAP experiment will be needing to experience ejaculation from the male mice twice, once to label the cells, and second during the reactivation. In order to avoid pregnancy during the first experience, we ovariectomized the female and controlled their hormonal conditions. This method has been used successfully in other sexual behavior studies (Yang et al., 2013, Ring., 1944.). This was described in P11. We have further demonstrated in Figure 1N-T that female mice were not ovariectomized and were under the natural estrus cycle showed similar suppression of sexual interaction after the completion of mating. The manuscript was updated to discuss that the behavior change after mating completion is not dependent on the ovary.

(11) Overall, the paper lacks references to relevant prior studies. For example, many studies have been reported over the past 2-3 decades about the effects of female rodent sexual behavior on activation in the brain and the effects of different vaginocervical stimulation on pregnancy and fertility. It is absolutely the case that much remains unknown about the complex neural circuitries that control behavior during the post-ejaculatory interval and sexual satiety in both male and female rodents, but studies have indicated roles for hypothalamic areas, bed nucleus of the stria terminals, ventral tegmental area, posterior thalamus, and prefrontal cortex. Hence, the current introduction and discussion do not adequately summarize or acknowledge these prior investigations and therefore place these new findings in the context of what was previously known.

We appreciate the comment and added references to P2 L65, P8 L355-357 to discuss existing literature about c-Fos mapping analysis after ejaculation or genital stimulation in female rats.

(12) Finally, sample sizes appear to be modest, ranging n=4-8 (except n=14 in the completion group in Figure S7) and vary between groups within and between studies. Please explain in the methods section how sample sizes were pre-determined and acknowledge if studies may have potentially been underpowered.

The sample size for behavior experiments in this study were n = 6-9. This was predetermined based on previous studies examining female sexual behavior (Ishii et al. 2017, Liu et al. 2022, Yin et al. 2022). To further examine the number of animals required for our behavioral experiments, we pooled data used in this study and conducted a power analysis (n = 111 pooled data, control n = 94, stim n = 17). We conducted a power analysis using the variance calculated from pooled average time in isolation zone. These data were pooled from control animals in each experiment (eg. animals with GFP control virus injected, saline injected, etc.). The average time in isolation zone in the after ejaculation or after reactivating the completion cells was 420 ± 210 seconds, and 49 ± 91 seconds in the control group (mean ± s.d.). Within this population, we found that 5 animals were sufficient to detect the difference (p < 0.05, power = 0.8) in Students t-test. We have added this explanation in the supplemental experimental procedure, page P18, line 817-827.

**Reviewer #3 (Recommendations For The Authors):**
The authors should discuss the fact that the FosTRAP2 strategy labels neurons activated 3 hours before the 4-OHT injection. As the manuscript is written, it seems to suggest that the 4-OHT injection given following mating completion only labeled neurons activated during mating completion. This is very misleading. I respect the amount of work and rigor that went into these experiments. The single-cell imaging, implementation of the FosTRAP strategy, and behavioral analysis are all well executed. Novel insights into the neural regulation of female sexual drive can be gleaned from the neural imaging experiments. Unfortunately, the limitations of the FosTRAP strategy make those studies very difficult to interpret, and therefore, a more candid discussion and re-interpretation of the data from the FosTRAP experiments is needed.

We appreciate the reviewers comments and concerns about the TRAP method.

First, we agree that the FosTRAP method does not have the sensitivity to separate ensembles that happen within a short time window. From our preliminary results, we have observed that the cells that inject 4-OHT after mating completion induce more tdTomato cells in the MPN than injection after appetitive behavior or consummatory behavior (Author response image 1).

To further compare the difference between the “consummatory” and “completion” ensemble, we included an additional cohort where we TRAP cells responding to consummatory behavior. This cohort is added to Figure 2, 6, S3, S4, S9, S10 and S11. From the whole brain mapping of TRAP cells, we found that many hypothalamic and extended amygdala areas including the medial preoptic area, and the bed nucleus of stria terminalis were shown to have significantly larger tdTomato+ cell density in the completion group than in the appetitive group while there was a tendency that the consummatory group also had larger cell density than the appetitive group. In the Gq-DREADD experiment, we found that the Completion-hM3Dq group but not the Consummatory-hM3Dq group showed the reduction of sexual motivation of the female mouse in the self-paced mating assay (Figure 6). The Completion-hM3Dq group but not the Consummatory-hM3Dq group also showed significantly low intromission events and tended to show lower receptivity in the home cage mating assay (Figure S10). Furthermore, post-hoc histological analysis also showed that the c-Fos+ and TRAP labeled cells in the MPN tended to be the larger in the Completion-hM3Dq group than in the Consummatory-hM3Dq group (Figure S9). These results, together with the *in vivo* Calcium imaging experiments in Figure 3, 4 and 5, suggests that the MPN contains male-ejaculation responsive cells that are distinct with the male-mounting responsive cells and that they are sufficient to suppress female sexual motivation.

However, it is true that with the current state of mouse genetic tools, we do not have any methods with higher time accuracy. We have discussed the limitations of FosTRAP method regarding its low time sensitivity in the Discussion section.

**Editor notes:**
Should you choose to revise your manuscript, please include full statistical reporting in the main text including test statistic, degrees of freedom, an exact P value.

Thank you for the comment. The statistical values were added to the manuscript.